

# Unveiling the Perth Canyon and its deep-water faunas

Julie A. Trotter[1,2], Charitha Pattiaratchi[1], Paolo Montagna[3,4], Marco Taviani[3], James Falter[1,2], Ron Thresher[5], Andrew Hosie[6], David Haig[2], Federica Foglini[3], Quan Hua[7], Malcolm T. McCulloch[1,8]

[1]Oceans Graduate School and UWA Oceans Institute, The University of Western Australia, Perth 6009, Australia
[2]School of Earth Sciences, The University of Western Australia, Perth 6009, Australia
[3]Istituto di Scienze Marine (ISMAR), Consiglio Nazionale delle Ricerche, 40129, Italy
[4]Laboratoire des Sciences du Climat et de l'Environnement, Gif-sur-Yvette, 91198, France
[5]Marine and Atmospheric Research, Commonwealth Scientific and Industrial Research Organization, Hobart, 7004, Australia
[6]Western Australian Museum, Welshpool, 6106, Australia
[7]Australian Nuclear Science and Technology Organisation, Kirrawee DC, NSW 2232, Australia
[8]ARC Centre of Excellence in Coral Reef Studies, The University of Western Australia, Perth 6009, Australia

*Correspondence to*: Julie Trotter (julie.trotter@uwa.edu.au)

**Abstract.** The Perth Canyon is a prominent submarine valley system in the southeast Indian Ocean that incises the southwest Australian continental shelf. It is characterised by two main steep-sided valleys forming a V-shaped configuration that extend from a depth of ~600 m to the abyssal plain at ~4000 m. Despite its prominence and location of only ~27 nautical miles (50 km) offshore, this study represents the first ROV-based exploration of the canyon and its inhabitants. ROV surveys revealed quiescent environments, the structure essentially representing a 'fossil canyon' system with localised occurrences of significant mega- and macrobenthos in the depth range of ~680 to ~1800 m. The patchy distribution of canyon life comprised corals, sponges, molluscs, echinoderms, crustaceans, brachiopods, and worms, as well as plankton and nekton (various fish species) especially near benthic communities. High definition video surveys and biomass sampling were complemented by ship-based multi-beam bathymetry, and seawater CTD profiling and chemical analyses.

ROV transects were conducted at six geomorphologically distinct locations, from the head to the mouth of the canyon and on the northern shelf plateau. The dives traversed the generally featureless muddy canyon floor, along near vertical walls, and onto the canyon rim. ROV imaging revealed typically massive and well-bedded sedimentary units that are variably lithified and mostly friable. Biostratigraphic and palaeoecological analysis of foraminifers from rock and sediment samples (~700 to 1600 m) indicate that they were deposited from the Late Paleocene to Early Oligocene within upper-middle bathyal (~200 to ~700 m) water depths, thus implying that significant subsidence has occurred. Strontium isotope ($^{87}Sr/^{86}Sr$) dating also suggests the presence of Early Miocene sediments at the shallower sites.

Settlement of large benthic sessile organisms is largely limited to indurated substrates mostly along the canyon walls. Corals were specifically targeted, with solitary (*Desmophyllum dianthus*, *Caryophyllia* sp., *Vaughanella* sp., and *Polymyces* sp.) and colonial (*Solenosmilia variabilis*) scleractinians found sporadically distributed along the walls and beneath overhangs in the deeper canyon valleys as well as along the canyon rims. Gorgonian, bamboo, and proteinaceous corals were also present with noticeable patches of live *Corallium* hosting a diverse community of organisms. Extensive coral graveyards were discovered



between ~690-720 m and 1560-1790 m comprising colonial (*S. variabilis*) and solitary (*D. dianthus*) scleractinians, which had flourished during the last ice age between ~18 ka to 33 ka (BP).

Faunal sampling (674 m to 1815 m) spans the intermediate and deep waters, which were identified as Antarctic Intermediate Water and Upper Circumpolar Deep Water, with temperatures of ~2.5 to ~6°C. The carbonate chemistry of those water depths

show supersaturation ($\Omega_{calc}$~1.3 to 2.2) with respect to calcite, but mild saturation to undersaturation ($\Omega_{arag}$ ~0.8 to 1.4) with respect to aragonite. Notably, some scleractinians inhabit depths below the aragonite saturation horizon (~1000 m). Depth profile measurements of $\delta^{13}C$ and nuclear bomb produced $\Delta^{14}C$ show decreases within the upper canyon waters of up to ~0.8‰ (< 800 m) and 95‰ (< 500 m) respectively, relative to measurements taken nearby in 1978, thereby reflecting the ingress of anthropogenic carbon into upper intermediate waters. Thus, the canyon waters and its inhabitants are already being subject to

the influences of $CO_2$ induced climate change and ocean acidification.

## 1 Introduction

Submarine canyons are a key feature of continental-shelf margins and depending on their mode of formation, geomorphology, and location on the shelf, can provide a variety of important functions (Huang et al., 2014). While some canyon systems represent the sub-sea extension of large river systems, transporting terrestrial sediments across the shelf to the deep-sea ocean

basins, many submarine canyon systems have no direct landward connection thus being largely isolated from terrestrial inputs. This latter class of canyons may be confined either to the more steeply dipping edge of the shelf-slope, or occasionally deeply incise the main shelf margin itself as in the case of the Perth Canyon (Harris and Whiteway, 2011; Huang et al., 2014). Previous studies of the Perth Canyon have been confined to limited dredge sampling and bathymetry (Marshall et al., 1989 and Heap et al., 2008), as well as opportunistic CTD ship-based characterisation of its deeper water masses (Rennie et al.,

2009; Woo and Pattiaratchi, 2008). Deep, shelf-incising systems like the Perth Canyon can, depending on the local oceanography (Pattiaratchi et al., 2011), provide a pathway for upwelling deep, nutrient rich waters onto an otherwise oligotrophic continental shelf (Rennie et al., 2009). Indeed, the steep-walled valleys of the Perth Canyon, which extend to the abyssal plains of the Indian Ocean, provide a conduit for upwelled nutrient-rich waters onto the shelf thereby resulting in a highly productive zone along the canyon rim and flanks that is used as a seasonal feeding-ground by migratory mega-fauna,

such as sharks and whales (Rennie et al., 2009). Thus, this configuration of steep-sided valleys with access to nutrient-rich deep waters would likely provide a variety of habitats for deep-sea sessile fauna, such as cold-water corals. Although relatively isolated, the unique deep-sea communities inhabiting canyons are nevertheless vulnerable to anthropogenic warming and ocean acidification, as well as direct human impacts from fisheries, land-based pollution, shipping activities, and oil and gas operations (Fernandez-Arcaya et al., 2017). With rising levels of anthropogenic $CO_2$, the carbonate saturation

horizon in the oceans is expected to rise by ~100 m to ~300 m by the year 2100 (Orr et al., 2005). This is likely to have severe consequences (Roberts et al., 2006), especially on deep-sea biota that precipitate carbonate skeletons (e.g. corals, molluscs, echinoderms) which are thereby subject to the combined effects of reduced calcification as carbonate saturation state decreases,



and increased dissolution of skeletons as ambient canyon waters become more corrosive (Thresher et al. 2011, McCulloch et al., 2012a).

Being near the south-eastern Indian Ocean margin (115°E, 32°S), the Perth Canyon waters are mostly sourced from southward coastal tropical surface waters, and north flowing intermediate and deep waters originating from Antarctica, and thereby

represents a prime location for studying changes in the ocean-climate system and their impacts on deep-sea ecosystems. A key aim of this expedition was to undertake the first remotely operated vehicle (ROV) based study and strategic collection of deep-sea corals from the Perth Canyon, to provide a reference 'baseline' of the changing carbonate chemistry of the deep-sea habitats that occupy this canyon system (McCulloch et al., 2017).

## 1.1 Formation and geological setting of the Perth Canyon

Prior studies (Marshall et al., 1989; Shafik, 1991; Heap et al., 2005) of the Perth Canyon have characterised its morphology and geological setting from lithological and biostratigraphical assessments of dredge, core, and sediment collections collected from the canyon walls, slopes, and seafloor. Its formation, however, is somewhat enigmatic, with no clear consensus on its origin and subject to controversy given the lack of connectivity to the nearby Swan River. The canyon is therefore surprisingly poorly understood despite being located only ~27 nautical miles (50 km) offshore Perth, the main population centre and capital

city of Western Australia.

The Perth Canyon resides within the Vlaming sub-basin, a pull-apart basin (Marshall et al., 1989) that formed during the separation of greater India from the western margin of the Australian continent, which concluded by the Early Cretaceous (~134 Ma to 137 Ma). Structural analysis by Marshall et al. (1989) indicates that the canyon is likely delineated by these pre-existing likely rift related fault structures given the now passive nature of the western Australian continental margin. Both the

NE-SW and SE-NW trending tributaries of the Perth Canyon appear to be aligned with several transfer faults and the head of the canyon is defined by small WNW trending graben (Marshall et al., 1989).

It has also been suggested that the canyon's incision of the continental slope is related to sub-aerial palaeo-drainage patterns carved by the adjacent Swan River when it formed part of a larger river system (von der Borch, 1968; Playford et al., 1976; Seddon, 2004). Although the canyon lies adjacent to the mouth of the Swan River there is, however, no Cenozoic

geomorphological evidence for any direct connectivity between the Swan River and the head of the submarine canyon, which is now ~600 m below sea level. This is also the case during Quaternary periods of low sea level stands (~120 m) where there is little evidence for the seaward migration of the Swan River mouth that could account for the extent of shelf-edge incision represented by the canyon. This does not preclude the possibility that a much more dynamic palaeo-Swan River drainage system provided precursor structures for the Perth Canyon, the initial uplift and down-cutting associated with either the late

Jurassic-Early Cretaceous rifting of India from the western Australian margin, and/or the Late Cretaceous (mid-Santonian) breakup of Antarctic from the southern Australian margin (Exon et al., 2005).

Although the Perth Canyon is one of the few examples of significant incision into the Australian shelf-edge margin (Heap et al., 2008), its formation thus appears to be a result of tectonic controls along pre-existing basin structures, together with more





recent (Tertiary) over steepening of shelf-edge sediments inducing submarine mass flow deposits and large-scale canyon cutting. Given that the western Australian shelf edge is part of a passive margin, the timing of this incision event or events remain poorly constrained except that they cut the youngest sediments (i.e. Miocene). Nevertheless, it is clear that, given the mass of sediment still preserved on the abyssal plains, these submarine mass flow events played a significant role in defining

the present-day geomorphology of the canyon.

Dredge samples recovered from the canyon walls and continental slope (~650 and 2400 m) during the 1988 R/V Rig Seismic expedition (BMR Cruise 80) comprised variably lithified limestones, shale/mudstones, and sandstones (Marshall et al., 1989). Their foraminiferal and nannofossil biostratigraphic ages range from Late Cretaceous to Early Miocene (Marshall et al., 1998; Shafik, 1991), with palynomorph ages ranging between the Middle Cretaceous and Permian, with the Permian and Triassic

samples considered as primary rather than reworked Mesozoic fossils (Marshall et al., 1989). The main stratigraphic units in the Perth Canyon have been related to the Upper Paleocene-Lower Eocene Kings Park Formation, Middle Eocene Porpoise Bay Formation, Upper Eocene Challenger Formation, and the Lower-Middle Miocene Stark Bay Formation (Marshall et al., 1989; Shafik, 1991; Fig. 5). The depositional environments have been interpreted as varying from shallow marine to estuarine facies associated with the Swan River palaeo-drainage system (Playford et al., 1976; Quilty, 1978). A study of the nearby

Challenger 1 drill core (WAPET 1975 oil exploration well), collected approximately 50 km south of the Perth Canyon, reported dominantly carbonate lithologies with planktonic foraminifera indicating age ranges from Late Palaeocene to Late Miocene (Quilty, 1978) thus similar to samples from the Perth Canyon.

Although not a major aim of the R/V Falkor cruise, new information on the geology of the canyon and surrounding shelf is provided by the first video footage of strata together with observations recorded in event logs taken during each dive. Some

additional geological data have been gathered from spot sampling of rocks and sediments from the substrate, canyon walls, and plateau, while collecting faunal samples by the ROV. An extensive multi-beam Sonar mapping programme undertaken during this cruise has further characterised the canyon morphology that includes the surrounding plateau, thus extending beyond the earlier mapping surveys (Fig. 1). A suite of bathymetry maps were also generated post-cruise at different spatial resolutions, which capture both the overall canyon system as well as localised topographic features at each of the ROV dive

location where sampling was undertaken.

## 2 Materials and Methods

Seawater and faunal samples were collected from the Perth Canyon during the research cruise FK150301 in early austral autumn from March 1st to 12th 2015. This cruise was undertaken using the R/V Falkor, the research vessel of the philanthropic Schmidt Ocean Institute, which was equipped with a CTD Rosette for seawater sampling and multi-beam sonars for high-

resolution bathymetric mapping. A Sub-Atlantic Comanche 2000m ROV operated by Neptune Marine Services, was employed for the strategic sampling of key macrobenthos from six dive sites (comprising nine dives). A comprehensive list of all samples collected during the cruise and complementary datasets are reported in the SOI Cruise FK150301 Final Report (McCulloch et



al., 2017, see: https://schmidtocean.org/cruise/perth-canyon-first-deep-exploration/). Note that the dive numbers relate to the nine survey and faunal collection dives (see McCulloch et al., 2017) rather than the SOI dive files, some of which are offset by one due to SOI D07 being a USBL recovery dive. Specimens collected during the cruise have been deposited with the Western Australian Museum and samples identified for geochemical analysis are held at the University of Western Australia.

## 2.1 Multibeam bathymetric analysis

High-resolution bathymetric data were acquired using Kongsberg EM 302 and 710 multi-beam echo sounders. A comprehensive mapping program initially focused near the six sample collection sites, ROV dive sites A–F (see Sect. 3.1.1), and then expanded to fill the remaining unmapped zones across the region (McCulloch et al., 2017). Post-cruise, bathymetry maps of the canyon and all dive sites were generated at different spatial resolutions (see Sect. 3.1.1). The multi-beam echo-sounder data was processed using CARIS HIPS and SIPS 7.0 software. After loading raw data and applying tide corrections, the data were 'cleaned' using the Swath Editor that consists of interactively selecting and rejecting soundings as well as filtering functions, which automatically detected and rejected outliers. Following data correction and cleaning, a digital terrain model (DTM) was generated at 20 m resolution for the entire canyon, and at 10 m resolution for the ROV dive areas (see Sect. 3.3). The DTMs were exported in Ascii ESRI format and analysed with ArcGIS 10.2. The ArcGIS Spatial Analyst tool was applied to derive the hill shade from the DTM with a vertical exaggeration of 1.5.

## 2.2 Seawater analyses

Water samples were collected between 15 and 2000 m from 5 of the dive sites (A–C, E–F) using 12L Niskin bottles mounted on a Rosette system equipped with a Seabird SBE 9plus CTD, with a SBE 43 dissolved oxygen sensor and Wet Labs ECO-FLNTU. Water column profiles are based on 59 measurements of temperature (T), salinity (S), dissolved oxygen (DO), chlorophyll $a$ concentration ($\Sigma$chl$a$, via water column fluorescence) total alkalinity (TA), dissolved inorganic carbon (DIC), and dissolved inorganic nutrients, together with calculations to determine the full suite of carbonate chemistry parameters. Seawater TA was measured on board by single-point titration based on spectrophotometric measurement of the end-point pH (Yao and Byrne, 1998). Seawater dissolved inorganic carbon (DIC) was measured on board using an Apollo SciTech Dissolved Inorganic Carbon analyser. Dissolved inorganic nutrients, including ammonium ($NH_4^+$), nitrate+nitrite (commonly referred to as just 'nitrate' or $NO_3^-$), phosphate ($HPO_4^{2-}$), and silica ($Si(OH)_4$) were analysed at the University of Western Australia on a Lachat autoanalyser using standard spectrophotometric methods. Seawater pH, $p$CO$_2$, calcite saturation state ($\Omega_{calcite}$), and aragonite saturation state ($\Omega_{aragonite}$) were calculated post-cruise using MATLAB version 1.1 of the CO2SYS software (van Heuven et al., 2011).

Stable C and O isotope measurements of 30 seawater samples (sites A, B, E) were undertaken in the West Australian Biogeochemistry Centre at The University of Western Australia. The stable isotope compositions of oxygen and hydrogen were analysed using an Isotopic Liquid Water Analyser Picarro L1115-i. Each sample was analysed six times and then the first four results were discarded in order to minimize any instrument memory effect. The $\delta^2$H and $\delta^{18}$O raw values of samples were





normalized to the VSMOW (Vienna Standard Mean Ocean Water) scale, based on three laboratory standards, each replicated twice and reported in per mil (‰) following the principles of the three-point normalization (Skrzypek, 2013). All laboratory standards were calibrated against international reference materials that determine the VSMOW-SLAP scale (Coplen, 1996), provided by the International Atomic Energy Agency (for VSMOW2 $\delta^2H$ and $\delta^{18}O$ of equal 0‰ and for SLAP equal $\delta^2H$ = -

428.0‰ and $\delta^{18}O$ = -55.50‰). The long-term analytical uncertainty (1σ) was determined as <1.0‰ for $\delta^2H$ and <0.1‰ for $\delta^{18}O$ (Skrzypek and Ford 2014). The stable carbon isotope composition of dissolved inorganic carbon (DIC) was analysed using a Thermo-Fisher GasBench II coupled with Delta XL Isotope Ratio Mass Spectrometer. Prior water sample injection, 12mL vials with 0.1 mL of 100% phosphoric acid were flushed with ultra-high purity helium (99.999%), and then reacted at 25°C for 24 hours (Paul and Skrzypek 2006). The water volumes of samples were adjusted accordingly in order to match the

peak heights of reference materials and avoid the linearity effect. All results were expressed using the standard delta-notation ($\delta^{13}C$) and were reported in per mil (‰) after normalization to Vienna Pee Dee Belemnite isotope scale [VPDB]. The multi-point normalization was based on three international standards NBS18, NBS19 and L-SVEC for $\delta^{13}C$, each replicated twice (Skrzypek, 2013). The analytical uncertainty was ≤±0.1‰ (1σ).

Radiocarbon ($^{14}C$) analyses of 20 seawater samples collected from sites A–C and E–F were undertaken at the Australian

Nuclear Science and Technology Organisation (ANSTO). Dissolved inorganic carbon of seawater (40-60 mL) was stripped out as $CO_2$ by acidifying the samples with 85% $H_3PO_4$ (5 mL), which was carried out in a custom-built extraction line by sparging the acidified water with He gas for 15 minutes. The gas was recirculated, passing through two dry ice/ethanol traps to remove water, and a $LN_2$ trap to condense the sample $CO_2$. Following removal of the He and other incondensable gasses, the $CO_2$ was converted to graphite using the $H_2$/Fe method (Hua et al., 2001). A portion of graphite was used for the

determination of $\delta^{13}C$ for isotopic fractionation correction using a Micromass IsoPrime elemental analyzer/isotope ratio mass spectrometer (EA/IRMS). Accelerator mass spectrometry (AMS) $^{14}C$ analysis was conducted using the STAR facility at ANSTO (Fink et al., 2004), with a typical analytical precision of better than 0.35% (1σ). Oxalic acid I (HOx-I) was used as the primary standard for AMS $^{14}C$ determination, and oxalic acid II (HOx-II) and IAEA-C7 reference material were employed as secondary standards. The AMS results are presented in $\Delta^{14}C$, the per mil deviation from the absolute radiocarbon standard

(Stuiver and Polach, 1977).

CTD data were visualized using the Ocean Data View software (version 4.7.10), and all graphs of seawater parameters discussed above have been generated using this programme.

## 2.3 U-Th dating

Absolute ages were determined by the uranium-thorium ($^{238}U$-$^{230}Th$) series decay technique for 28 fossil coral specimens

(n=31) collected at two ROV dive sites spanning a depth range of 691 to 1788 m. Subsamples were extracted from the fossil skeletons of *Desmophyllum dianthus* and *Solenosmilia variabilis* using a dental drill with diamond-encrusted blades and burs. The subsamples were first mechanically cleaned with the drill then crushed in an agate mortar and pestle. Aliquots of ~20 to ~50 mg were lightly leached in dilute (0.1N HCl), dissolved in dilute HCl and spiked with enriched $^{229}Th$, $^{233}U$ and $^{236}U$ tracers.



Uranium and thorium were extracted from 3N $HNO_3$ sample solutions using Eichrom UTEVA ion exchange chromatography resin following the procedure of Douville et al. (2010). All U-Th measurements were carried out on a ThermoScientific Neptune Plus MC-ICPMS equipped with a jet interface and a desolvation system yielding an overall ion efficiency of 2-3%. The $^{238}U$ signal was collected using a $10^{10}$ Ω amplifier, allowing signal intensities for $^{238}U$ of >100V. U samples and bracketing

standard CRM 112-A were spiked with reference material IRMM-3636a to correct for mass bias and yield drift of the SEM. Additional corrections were applied for tailing and hydride generation. For Th determinations, $^{229}Th$-spiked samples were bracketed against IRMM-3636a, with specific attention given to maximise washout of $^{230}Th$ between samples. Using U as the bracketing standard for Th may introduce some uncertainty from differential mass bias behaviour; however, no suitable Th standard was available and errors from mass bias correction are generally smaller (<1‰) than errors from counting statistics

on $^{230}Th$ alone (~3-8‰). Activity ratios for $^{230}Th/^{238}U$ and $^{234}U/^{238}U$ were then calculated using the decay constants and reference values given in Cheng et al. (2013) and Jaffey et al. (1971), but calibrating the $[^{230}Th/^{238}U] = 1$ for Harwell Uraninite HU-1 assuming secular equilibrium, and using $\delta^{234}U$ = -38.5‰ for CRM112-A. Analytical uncertainty for $\delta^{234}U$ derived from repeated determinations of HU-1 was 0.8‰ (2σ; n=30), whereas errors on $[^{230}Th/^{238}U]$ strongly depend on $^{230}Th$ counting statistics and minor blank/washout corrections, and ranged from ~2.8‰ (2σ; n=30) for HU-1 to 8.3‰ (2σ; n=12) for very

young samples (e.g. CRM112-A). Total procedural blanks were typically on the order of ~30 attograms for $^{230}Th$ and ~12 femtograms for $^{238}U$, and are considered negligible. Ages were calculated through iterative age estimation (Ludwig and Titterington, 1994) and assuming the initial non-radiogenic $[^{230}Th/^{232}Th] = 2$ (±1) then calibrated to the year 1950 (i.e. BP). All samples were prepared and analysed in the Advanced Geochemical Facility for Indian Ocean Research at the University of Western Australia.

**2.4 Geological samples and biostratigraphy**

Several geological 'spot' samples were collected using the ROV robotic arm. Foraminiferal biostratigraphy, palaeo-bathymetry, and lithologies have been determined from a selection of nine geological samples from dive sites A and D–F (see Sect 3.1.2). Seven friable to partly friable samples were disaggregated in water and the washed sand fractions examined under a stereomicroscope. Indurated samples of wackestone and chert were examined in thin section. The samples investigated were

from depths of 1603 m (FPC-15 D05-S01), 1241 m (FPC-D05-S08), 1032 m (FPC-D07-S05), 746 m to 716 m (FPC-15 D09-S01, S02, S04) and 701 m (D08-S03). The biostratigraphy of these samples was determined from planktonic foraminiferal associations and the palaeo-bathymetry from benthic foraminiferal assemblages.

**2.5 Sr isotope dating**

The Sr isotopic compositions of sediments sampled from the basal holdfasts of eight deep-water corals together with seven

bedrock samples from five dive sites (A-B, D-F), were used to calculate Sr ages (see Sect. 3.2.1). Sediments extracted from coral holdfasts were from depths of 1472 m (Site E D05-S03, live *Vaughanella*), 936 m (Site B D04-S01, live and fossil *Desmophyllum* and fossil *Solenosmilia* sampled from the same colony), 739 m (Site F D09-S02, live *Polymyces*), 701 m (Site



A D08-S03, recently dead *Polymyces*), and 675 m (Site A D08-S05, live *Caryophyllia*). Sample collection D09-S2 was from a small boulder sitting on the substrate. Rock analyses were of subsamples from well-consolidated mudstone units sampled from the canyon wall at 1444 m (Site E D05-S04 mudstone) and 1032 m (Site D D07-S05), as well as soft, friable mudstone nodules collected from the shallower depths at 746 m (Site F D09-S01), 716 m (Site F D09-S04), and 701.3 m (Site A D08-

S3). Collection D09-S02 comprises loose debris that was scooped from the substrate. The very small size and preservation of the foraminifers made them unsuitable for isotope analysis, thus precluding comparative Sr analysis of the host rocks and microfossils that were used for biostratigraphy.

Strontium was separated from 10 mg carbonate aliquots by ion exchange chromatography using Eichrom Sr Resin. All Sr isotopic compositions were measured in solution using the ThermoScientific Neptune Plus housed in the Advanced

Geochemical Facility of Indian Ocean Research at the University of Western Australia. Averages of standard $^{87}Sr/^{86}Sr$ measurements from two analytical sessions gave NBS SRM987 = 0.710299 (n = 7) and 0.710283 (n = 4), with analytical uncertainties at 95% confidence limits of 0.000015 and 0.000004 respectively. The instrumental mass fractionation was corrected by using a stable isotopic $^{86}Sr/^{88}Sr$ of 0.1194 and an exponential law. $^{87}Sr/^{86}Sr$ measurements of the sample unknowns were normalised to 0.710248 to determine their Sr ages using the SIS Look-up Table v4:08/03 (McArthur et al., 2001).

## 3 Results

### 3.1 Bathymetry and geomorphology of the canyon

As described in earlier studies (von der Borch, 1968; Marshall et al., 1989; Heap et al., 2008), the canyon is characterised by two main tributaries forming a sinuous or V-shaped configuration, which initially trends SW from the canyon head, then sharply diverges to the NW towards the mouth of the canyon (Figs. 1, 2). A series of approximately right-angle bends mark

the major changes in the canyon's orientation at distances of ~10 km, 40 km, 50 km and ~100 km from the tip (the head's closest point to the coast). From the easternmost tip, the canyon's orientation at 0-10 km trends in a ~SE-WNW direction, at 10-40 km it runs NE-SW, at 40-50 km it trends roughly E-W, at 50-100 km it changes to a ~SE-NW direction, and at ~100-120 km the mouth follows a near westerly direction as it opens onto the abyssal ocean plain. A morphologically more complex zone occurs in the region ~40-50 km from the tip, where the canyon's floor narrows in parts to <1 km in width and its

orientation changes abruptly thus defining the major canyon bend (herein Site C: "Dog-Leg Canyon"). This main juncture at ~50 km is joined by a short southern arm, which is a ~15 km long trench with a similar morphology to the main (inshore) canyon head.

The walls of the canyon typically have ~30° to 40° gradients, although some parts are near vertical, such as at site C and E (herein Dog-Leg Canyon and Amphitheatre Waterfall respectively). These steep vertical walls sometimes ~500 m high were

key targets for ROV exploration during the Falkor cruise, as they were most likely to provide substrate suitable for aggregations of deep-water corals. The width of the canyon varies from roughly 2-5 km wide near the head, broadening to ~10 km in the



10-40 km region, narrowing again to ~5-8 km at Site C, and eventually expanding to ~20 km at the mouth as it forms a large submarine fan.

Notable geological features, such as tilted faulted blocks and subsurface channelling with infill near the southern (second) head (Marshall et al., 1989), as well as variably developed sand waves, slumped blocks, and sediments, are indicative of

sedimentary processes that have transported upper slope and shelf sediments into the canyon (Heap et al., 2008). Some of these structures as well as the extensive shallow terraces near the canyon walls and across the plateau are apparent in the high-resolution bathymetry maps and ROV video footage and stills obtained during the R/V Falkor cruise (also see https://schmidtocean.org/cruise/perth-canyon-first-deep-exploration/#data).

The nine survey and collection dives were undertaken at six geomorphologically distinct sites (A-F) around the canyon, at

each end of the two valley systems and the adjacent continental slope (Fig. 2). Two dive sites (A, B) are located at the head of the canyon in the east, two (C, D) are in the south near the sharp junction of the main canyon valleys, and two are in the northwest near the canyon mouth (E) and above the canyon on the upper northern plateau (F). The higher resolution bathymetry maps for each dive site (Fig. 3) show the local morphology as well as the ROV paths.

Dive sites A and B near the canyon head feature a series of broad terraces and represent two of the three shallowest dive sites.

Site A represents one of the two shallowest ROV traverses along which samples were collected (~800 to ~600 m) from the base and up part of the terrace wall. Opposite Site A, similar terrace structures occur at Site B where samples were collected (~1000 to ~600 m) along a slightly deeper traverse.

Closest to the narrow junction of the two major valleys in the southwest is dive Site C, where the canyon walls are near vertical with heights of ~500-600 m and represent the deepest and steepest traverse from which samples were collected (~1800 to

~1500 m). Directly opposite at dive Site D, the gradient and height of the canyon wall is comparatively shallower. Here, almost the full height of the canyon was surveyed and sampled by the ROV (~1200 to ~800 m).

At the end of the main northwest trending valley towards the mouth of the canyon, lies a deep, steep walled, large arcuate terrace, which represents dive Site E and the second deepest sampling range (~1700 to ~1200 m). This area has topography reminiscent of a terrestrial waterfall, and from the top of the wall a well-defined northeast trending scarp runs parallel to the

main (inshore) eastern valley, leading to the shallow northernmost dive Site F, where faunal were samples collected (~800 to ~600 m) across a shallow plateau.

### 3.2. Geological characteristics of dive sites

The rock lithologies comprising the canyon walls and substrates vary with depth and between dive sites. The canyon walls at the deeper sites (C and E) are characterised by well-bedded and massive rocks, whereas the shallow water sites (A, B, F) are

dominated by less consolidated, friable, thinly bedded units (Fig. 4; Supplement Figures S1 and S2). Muddy deposits occur throughout the canyon, sometime as large mounds near the substrate and as wide sediment aprons draping rocky outcrops and cliffs. A synopsis of the environments observed at each site is given below.



Site A (Glider Crash, dives 2 and 8) near the canyon head was surveyed between depths of 743 and 663 metres (Fig. 4a-c). The highly bioturbated modern muddy substrate with sparse benthos sharply transitioned to a coarse bedded vertical cliff-face with fissures and prismatic jointing. Changes in lithology during the dive comprised intervals of brecciated rock, interbeds of blocks with slump-like structures, mudstone and chalky layers of varying thickness, and thin nodular lenses. Thinly bedded

units show dissolution features and possible caliche, and the rocks sampled are soft and friable.

The nearby Site B (Derwent Wreck, dive 4), was traversed from 1092 to 911 metres (Fig. 4d-f). From the seabed, a steep muddy slope led to a mud-draped cliff with 'corniche-like' features, cavernous overhangs, muddy aprons, and well-bedded mudstone units. Thinly bedded units and fine sand/silt draping over bedded outcrops or forming mounds dominate this site. The rocks sampled are soft and friable.

Site C (Dog-Leg Canyon, dives 3 and 6), located at the tributary juncture, was the deepest survey that ranged between depths of 1821 to 1512 metres (Fig. 4g-i). The steep muddy seabed transitioned to a towering cliff, comprising massive blocky and jointed strata, and well-bedded consolidated units variably (sometimes steeply) dipping, their surfaces relatively smooth to coarsely textured (and striated) with crossbedding features. Frequent intervals of large muddy aprons occur between the cliff walls, with masses of fossil coral debris protruding through the sediments.

The nearby Site D (Glass Sponge Ridge, dive 7) was surveyed from 1210 to 834 metres (Fig. 4j-l). The highly bioturbated muddy substrate transitioned to a well-bedded cliff draped with mud. Layered mudstones, cherts, large collapsed blocks, steep cliffs (sometimes intensely bio-eroded) and well-defined beds, featured throughout the dive. Cherty layers and soft clay beds are common at the top of the dive.

The second deep survey was at Site E (Amphitheatre Waterfall, dive 5), located near the mouth of the canyon, which ranged

from 1728 to 1241 metres (Fig. 4m-o). The bioturbated muddy substrate transitioned to predominantly silty and mudstone bedded outcrops, fractured sandstone cliffs, massive blocky units and slumped blocks, with well-bedded strata and intermittent pebbly layers. Slump structures, wavy laminations, and brecciated units are apparent. Chert and soft clay beds are common near the top of the dive.

Site F (Two Rocks, dive 9), the shallow and northernmost site on the canyon plateau, was surveyed between the depths from

760 to 682 metres (Fig. 4p-r). The rocky seabed and adjacent cliffs featured rough eroded surfaces, and mud-draped slopes together with small blocks, pebbles, and rubble deposits. Strata are dominated by soft, friable, thinly bedded units, variably lithified with dissolution features. Extensive masses of fossil coral debris cover the fine sediment substrate at the top of the dive.

### 3.3. Geological history of the canyon

The biostratigraphic ages range from Late Paleocene to Early Oligocene, however some of the Sr isotope ages of samples from the shallow sites were anomalously younger, from Late Oligocene to Early Miocene. Notably, Oligocene sediments had not been reported from the Perth Basin prior to Shafik (1991), who determined a late Early Oligocene age for one of the dredge samples (80DR/022-4) previously collected from the Perth Canyon during BMR Cruise 80. Figure 5 summarises the





formations and biostratigraphy described in earlier studies together with the ages determined from the samples collected from the R/V Falkor cruise described below.

### 3.3.1 Biostratigraphy and palaeoecology

The nine geological samples represent a roughly 900 m sedimentary section that comprises three distinct facies ranging from

the Late Paleocene to Early Oligocene in age (Supplement Table S1). This section is significantly thicker than the condensed section (~240 m thick) of equivalent age determined from the Challenger 1 study by Quilty (1978). The palaeobathymetry determined from benthic foraminifers from the Falkor samples indicate upper bathyal to upper middle bathyal (200-700 m) depths throughout the Late Paleocene to Early Oligocene. The Oligocene samples were probably deposited at a depth similar to the present-day water depth at sites A and F, which are the shallowest dive sites. This interpretation and the great thickness

of the section compared to that of Challenger 1, suggests significant subsidence took place during the Eocene whereas little subsidence occurred during the Neogene. The facies identified are as follows:

(1) Late Paleocene: A wackestone was collected at 1602 m during the deepest sampling S1 of ROV dive 5 (D05-S01) at Site E (Amphitheatre Waterfall). The planktonic foraminiferal assemblage is dominated by species of *Subbotina*, *Globanomalina* is common, but *Acarinina* and *Morozovella* are rare. The co-occurrence of *Globanomalina pseudomenardii* and *Acarinina*

*soldadoensis* is consistent with planktonic foraminiferal Zone P4c hence a Late Paleocene age (~56-56.5 Ma) (Olsson et al., 1999). Benthic foraminifers are represented by abundant *Bulimina tuxpamensis* associated with *Spiroplectammina spectabilis*, *Oridorsalis umbonatus*, *Osangularia*, and *Pleurostomella*, which suggest bathyal zone depositional water depths between ~200 m and ~700 m, following the bathymetric distribution model of van Morkhoven et al. (1986). There is no evidence of down-slope movement of sediment from a neritic shelf environment thus assume the assemblage is *in situ*.

(2) Middle-Late Eocene: A wackestone was collected at 1241 m during sampling S8 of ROV dive 5 (D05 S08) at Site E. The foraminifera are poorly preserved in this sample that has incipient cement. The presence of *Globigerinatheka index* together with probable *G. subconglobata* and very rare minute *Acarinina* sp. with some resemblance to *A. topilensis* (Cushman), suggest a probable E10–E12 zonal placement hence Middle Eocene age (~40–44 Ma) (Pearson et al. 2006). The few benthic foraminifera recorded are consistent with an upper bathyal depositional water depth with no evidence of down-slope transport

from a neritic shelf environment.

An indurated wackestone was collected at 1032 m during collection 5 of ROV dive 7 (D07 S05) at Site D (Glass Sponge Ridge). This sample contains sporadic planktonic foraminifera, rare benthic species, and abundant sponge spicules. Although the planktonic foraminifera are difficult to identify in thin section the presence of *Globigerinatheka* indicates a Middle or Late Eocene age (zones E8–E16; ~34–48 Ma). The absence of distinct *Acarinina* or *Morozovelloides* may indicate an age no older

than zone E12, late Middle Eocene (no older than ~40 Ma), as does its position above sample FPC15-D05-S08. There is no evidence of down-slope movement of sediment from a neritic shelf environment.

(3) Early Oligocene: A number of wackestones collected during dives 8 and 9 from depths of 746 m to 701 m (D08-S03; D09-S01, S02 and S04) contain foraminifera of variable preservation that are seemingly the same age. The presence of *Catapsydrax*



*dissimils*, *C. unicavus*, *Globoturborotalita anguliofficinalis*, *G. ouachitaensis*, and *Dentoglobigerina tripartita* and the absence of *Globigerinatheka*, *Hantkenina*, and *Turborotalia cerroazulensis*, which are characteristic of the Late Eocene, as well as *Globigerinoides* that is characteristic of the Miocene, suggest that these samples are of Oligocene age. The lack of abundant *Globoturborotalita*, including the *G. ciperoensis* (Bolli) group of species, as well as *Globoquadrina binaiensis* (Koch) indicate

that the age is early Oligocene (zone O1–2; ~31–33 Ma). The benthic foraminiferal assemblage is consistent with an upper bathyal to upper middle bathyal water depth within the range 200-700 m (following van Morkhoven et al., 1986). There is no evidence of down-slope movement of sediment from a neritic shelf environment.

### 3.3.2 Strontium isotope ages

The Sr isotopic compositions determined from a number of hand samples and remnant host rock attached to some coral

holdfasts have provided some additional insights into the depositional ages of canyon lithologies (Supplement Table S1). The deepest samples, from 1472 m (D05-S03, coral holdfast) and 1444 m (D05-S04, mudstone) collected from Site E (Amphitheatre Waterfall), gave the lowest $^{87}Sr/^{86}Sr$ ratios (0.707775 - 0.707774) which equate to a broad range of ages (55.6-35.0 Ma) throughout most of the Eocene. For this age interval, the Sr isotope seawater curve is non-linear, thus the oscillations between 0.707702 and 0.707782 yield multiple age assignments for ratios within that range. These Sr ages are within the age

range determined by foraminiferal biostratigraphy that constrain the ages of samples collected at the bottom (D05-S01, Late Paleocene) and top (D05-S08, Late Eocene) of this dive. A sample from an intermediate depth of 1032 m (D07-S05, mudstone) at Site D, also yielded low $^{87}Sr/^{86}Sr$ ratios (0.707781, 0.707794) and hence multiple possible Sr ages (45.1-35.3 Ma) between the Middle to Late Eocene, thus consistent with the samples from >1000 m at site E. The Middle Eocene biostratigraphic age of forams from this rock hand specimen (D07-S05) suggests that the Sr age is between 43.9 and 45.1 Ma.

Sr ages of latest Oligocene and Early Miocene were determined for the remaining samples, which comprised sediments extracted from coral holdfasts, as well as collections of loose soft, friable, mudstone and siltstone nodules from shallower depths (936-675 m) near the head of the canyon at Site A and B, and on the canyon plateau in the northwest at Site F. For this time interval, the gradient of the Sr seawater curve is unidirectional and relatively steep, thus single ages with small errors can be calculated. From 936 m at Site B on the eastern canyon wall, samples from D04-S01 yielded consistent Early Miocene Sr

ages (20.3, 20.1, and 19.8 Ma) for sediments extracted from the holdfasts of fossil and live-caught corals. There are no biostratigraphic ages for samples from this location. Early Miocene ages (22.0 to 17.0 Ma) were also determined from substrate debris collected at Site F ~35 km to the northwest, from D09-S01 (mudstone, 21.9 Ma) at 746 m, from D09-S2 (coral holdfast, 17.4 and 17.0 Ma) at 739 m, and from D09-S04 (mudstone, 21.9 and 22.02 Ma; indurated pre-hardground, 20.5 Ma) at 716 m water depths. At a slightly shallower depth of 701 m at Site A on the western wall near the canyon head, three different samples

(siltstone nodule, mudstone nodule, coral holdfast sediment) from collection D08-S03 yielded similar Sr ages of Late Oligocene (23.9 Ma) and Early Miocene (22.4 and 20.1 Ma) respectively. From the shallowest depth of 675 m at this site, D08-S05 (coral holdfast sediment) yielded an age of 17.6 Ma.





The $^{87}Sr/^{86}Sr$ measurements therefore range between 0.707774 and 0.708660, which equate to Sr ages of 55.6 to 17.0 Ma thus spanning the Early Eocene to Early Miocene. However, there are apparent discrepancies between the Early Oligocene biostratigraphic ages and late Oligocene-Early Miocene Sr ages for samples scooped from the substrate at the shallower dive sites at the canyon head and plateau (A and F). The likely causes either invoke diagenetic processes or mixed sampling at these

sites, the former explanation having merit given their very soft and friable nature together with possible dissolution and weathering features. Nevertheless, we note that the Sr ratios are internally consistent between all samples analysed and that they were collected as loose debris and/or from fallen blocks resting on the substrate at the base of escarpments (see ROV tracks of Fig. 3). The Falkor cruise videos also show that the thick sequence of shallow mud-siltstone deposits incorporate what appear to be distinct intervals of small to large chert clasts and boulders, which occur across the canyon plateau at sites

A to F, these extensive sequences having brecciated characteristics potentially incorporating mixed sources and ages of sediments. Notably, Quilty (1978) inferred large masses of chert from high-velocity peaks in the Sonic Velocity logs of Eocene and Upper Miocene sequences but at the much shallower and condensed Challenger 1 Well site located farther south. Early Miocene foraminifer and nannofossil ages have been reported previously from dredge samples (80DR/007-1, -2, -3) collected from the Perth Canyon during BMR Cruise 80 cruise at 650-850 m on the top of the eastern plateau, southwest of Falkor cruise

Site B, which were interpreted to be from the Stark Formation (Marshall et al., 1998; Shafik, 1991).

### 3.4 Physical and chemical oceanography

### 3.4.1 Hydrography of surface waters

Strong vertical stratification and current shear has been observed at 300-350 m depth at the interface between the southerly flowing Leeuwin Current and the northward flowing Leeuwin Undercurrent. The canyon's influence on the Leeuwin Current

dynamics thus occurs within the shallower waters of the canyon head. However, the shape of the continental shelf near the canyon together with the separation of the Leeuwin Current from the shelf forms anticlockwise surface eddies during winter especially. Within the canyon, interactions between the Leeuwin Undercurrent and the canyon generates clockwise eddies, forming over a period of five to ten days, which then migrate offshore. Their clockwise rotation thus generates upwelling in the eddy centres. The recurrence of eddies within the canyon suggests that the canyon regulates the circulation, with several

circular eddies present both spatially and at different depths at any given time.

Oceanographic data collected during the cruise revealed the presence of an anti-clockwise eddy with the southward flowing Leeuwin Current located further offshore, west of 114.7ºE (Figs 6-7). The colder northward flowing Capes Current develops along the continental shelf due to strong southerly winds. Shelf water temperatures and time series of the currents are indicative of energetic diurnal currents within the water column which, together with strong wind driven upwelling (Fig. 8), develop

from the local sea breeze system (e.g. Mihanović et al., 2016).



### 3.4.2 Structure and chemical compositions of canyon waters

The waters in and around the Perth Canyon (sites A-C, E-F) are very similar in their physical structure and compositions (Fig. 9). Temperatures range from ~24°C in the shallow well-mixed surface waters to ~2°C in the deepest waters (2000 m). The salinity maximum of 35.8 occurs just below the well-mixed layer (100-250 m), with the minimum of 34.4 occurring at

mesopelagic depths (600-700 m). Dissolved oxygen ranges between ~250 µmol kg$^{-1}$ (300-550 m) and ~150 µmol kg$^{-1}$ (below 1000 m). Profiles of total chlorophyll exhibit a somewhat 'noisy' but persistent sub-surface maximum of 0.25 - 1.0 mg m$^{-3}$ that occurs between ~60 and 90 m at all locations, except the northernmost site (Site F) where the maximum of 0.36 mg m$^{-3}$ is much deeper at ~150 m (Fig. 9).

The key water masses from which the canyon waters originate have been identified based on their temperature, salinity, and

dissolved oxygen contents (Fig. 9), and in reference to prior oceanographic studies of this region (Woo et al., 2006; Rennie et al., 2009). The uppermost warm well-mixed Tropical Surface Waters (TSW ~50 m) is transported poleward by the Leeuwin Current, below which is the South Indian Central Water (SICW ~100-250 m) as indicated by maximal salinities (≥ 35.6). The subsurface chlorophyll maximum is generally within the transitory thermocline, separating the SICW and the well-mixed layer above it (50-100 m), but there are still measurable albeit low levels of chlorophyll in the SICW. The underlying north flowing

Subantarctic Mode Water (SAMW, > 250 to ~550 m) is characterized by maximal dissolved oxygen concentrations (> 250 µmol kg$^{-1}$). At intermediate water depths, north flowing waters sourced from the polar zone comprise Antarctic Intermediate Water (AAIW, ~650 to ~1000 m), which is clearly identified by consistent salinity minima (~34.4). A transition into colder and more saline deep waters together with oxygen minima (~150 µmol kg$^{-1}$) represents the presence of Upper Circumpolar Deep Water (UCDW, ~1000 to ~2000 m). The weak vertical gradients in temperature, salinity, and dissolved oxygen, provide

only approximate constraints for the depth boundaries of these waters.

The water column nutrients throughout the Perth Canyon are almost entirely planktonic in origin, as indicated by the ratio of total dissolved inorganic nitrogen (nitrate plus ammonium) to total dissolved inorganic phosphorus (SRP) being close to the idealized Redfield-Richards ratio (17.4 ±0.4 vs. 16). The amount of nutrients re-mineralized from planktonic organic matter increases with water age and depth, as indicated by the monotonic increase in nitrate plus nitrite and soluble reactive

phosphorus with depth (Fig. 10). However, there is no apparent vertical structure in ammonium concentrations, which are generally less than 0.5 µM throughout the water column, reflecting the tight coupling between rates of organic nitrogen remineralization and nitrification.

Water column profiles of total alkalinity (TA) and dissolved inorganic carbon (DIC) are similar across the canyon sites (Figs. 11). Profiles of TA appeared to have a similar shape as profiles of salinity, which reflects how much TA primarily depends

on the concentration of dissolved salts (including carbonate and bicarbonate) in the source seawater. Note that TA values at depth are 10-20 µmol kg$^{-1}$ higher than surface seawater values even though the salinity at depths greater than 500 m (≤ 34.6) are roughly one unit or more lower than at the surface (35.6). The higher TA despite the lower salinity reflects the cumulative dissolution of biogenic carbonate minerals within these deep and intermediate water masses over their lifetimes since



formation; material that is most likely pelagic in origin (Smith and Mackenzie, 2016). DIC increases more monotonically with depth owing to its elevation being dominated by the cumulative respiration of largely planktonic organic matter increasing with water age and depth, as well as to a comparatively modest amount of carbonate mineral dissolution.

The higher levels of DIC observed at depth are also partly influenced by the greater solubility of $CO_2$ gas during their original

formation in the polar region as colder surface waters prior to subduction. Given that DIC production by organic matter respiration dominates the non-conservative elevation of DIC within the deeper sections of the water column, there is a consummate decrease in pH and increase in $pCO_2$ with depth (Fig. 11). The rise of $pCO_2$ with depth more directly reflects the input of the respiratory carbon, however, the effect of this biogenic injection of inorganic carbon on vertical profiles of aqueous $pCO_2$ is somewhat muted by the increased solubility of $CO_2$ with depth as a function of decreasing temperature. The substantial

declines in pH with depth causes carbonate ion concentrations to decrease from ~220 µmol kg$^{-1}$ at the surface to 70-80 µmol kg$^{-1}$ at depths greater than 1000 m (data not shown) despite the 15-17% increase in DIC at depths greater than 1000 m relative to the surface.

Assuming changes in $Ca^{2+}$ concentrations and activities due to variations in salinity are comparatively minor, this implies that the ion activity product (IAP) of $CO_3^-$ and $Ca^{2+}$ also decreases by roughly two-thirds with depth.  In contrast, carbonate

mineral solubilities ($K_{sp}$) increase with depth due to lower temperature and, to a lesser extent, higher pressure. The combination of increasing IAP and decreasing $K_{sp}$ result in substantial monotonic declines of ~75% in both calcite and aragonite saturation states with depth relative to surface values (Fig. 11). Although all waters within the Perth Canyon and surrounds remain oversaturated with respect to calcite, waters deeper than ~1000 m are undersaturated with respect to aragonite. Thus, we can identify an aragonite saturation horizon within the Perth Canyon as being somewhere near 1000 m.  We expect that this

saturation horizon extends well beyond the Perth Canyon given the consistency between our observations of the vertical structure of waters in the Perth Canyon with observations made (Woo et al., 2006) further off the shelf.

The $\delta^{18}O$ profile (Fig. 12) shows a similar depth dependent trend to that of salinity, with the upper ~200 m waters being characterised by both high salinity Tropical Seawater (TSW) and South Indian Central Waters (SICW) as well as high $\delta^{18}O$, a consequence of the tropical source and evaporative history of these upper water masses. This is confirmed by the good

correlation between salinity and $\delta^{18}O$ (Fig. 12) with the high salinity end-member represented by the TSW and SICW waters. The hydrogen isotope profile (Fig. 12) also shows a similar pattern as the $\delta^{18}O$, thus consistent with the higher salinity of the upper ~200 to 300 m water masses having elevated $\delta^{2}H$.

### 3.4.3 Carbon isotope compositions and anthropogenic effects

The $\delta^{13}C$ composition initially decreases in depth within the uppermost ~200 m (~0.8 to 0.6‰), then rapidly increases to a

maximum at ~420 m (~1.2‰), before decreasing asymptotically to ~0.4 from ~1200 m (Fig. 13). This general profile is similar to those reported previously by Kroopnick et al (1985) for the eastern Indian Ocean, as well as Sonnerup et al (2000) and Quay et al (2003) for the central subtropical (~20°S) Indian Ocean waters collected during the GEOSECS (1978) and WOCE (1995)





cruises, these locations being further offshore and north of the Perth Canyon. Comparison of prior $\delta^{13}$C measurements closest to the Perth Canyon, undertaken in 1978 (GEOSECS station 436), 1995 (WOCE station 435), and 2009 (US CLIVAR/CO$_2$ station 189), show continued depletion of $^{13}$C within the upper ~800 m, the 2015 Perth Canyon data having the greatest negative offset of up to ~0.8‰ (at ~200 m) from the 1978 profile (Fig. 13). This decrease in $\delta^{13}$C composition reflects the gradual

ingress of low $\delta^{13}$C (-28‰, Andres et al., 1996) fossil fuel carbon into surface and upper intermediate water depths over this 37 year period (1978-2015). Between the years 1978 to 2015, the overall rate of change in seawater $\delta^{13}$C within the Perth Canyon region is -0.23‰ per decade. Similar mean decadal declines in $^{13}$C have been reported previously for the Indian Ocean and Atlantic subtropics (0.18 to 0.25‰) with slightly lower means given for the global ocean (Bacastow et al., 1996; Gruber et al., 1999; Sonnerup et al., 1999; Kortzinger et al., 2003; Quay et al., 2003; Eide et al., 2017).

Seawater radiocarbon ($\Delta^{14}$C) compositions from the Perth Canyon range from 42.9 to -191.2 from surface to deep waters (Fig. 13), and show a similar depth trend to that of existing profiles for the region determined in 1978 (GEOSECS station 436), 1995 (WOCE station 435), and 2009 (US CLIVAR/CO$_2$ station 189). These profiles systematically diverge within the upper ~500 m reflecting a decrease of ~ 95‰ from 1978 to 2015 in this region. This systematic decrease in upper ocean $^{14}$C reflects the gradual dilution of nuclear bomb carbon, produced in the late 1950s and early 1960s, together with $^{14}$C depleted

anthropogenic CO$_2$ released into the atmosphere from fossil fuel burning. Recognising the minor difference in site locations between past oceanographic surveys and this study of Perth Canyon waters, the rate in $^{14}$C decline in the upper waters is on average about 28‰ per decade between 1978 and 2015, however, the decadal rate has effectively halved between the two collection intervals, from 40‰ (1978-1995) to 20‰ (1995-2015). For comparison, the decline in atmospheric $^{14}$C (Orr et al., 2017; Turnbull et al., 2017) is about three times faster than the mean rate of decline in seawater $^{14}$C in this region, being only

one third of the atmospheric rate of decline (~80‰ per decade) for that period. The penetration depth of $^{14}$C (~500 m) is less than that of $^{13}$C (~800 m), which is consistent with relative penetration depths modelled for mean anthropogenic CO$_2$, $^{13}$C, and $^{14}$C and their relative time histories (Quay et al., 2003). Below the upper ~650 m, the radiocarbon content decreases from -25.6‰ (±2.7, 1σ) to -191.2‰ (±2.6, 1σ) between 651 m and 2000 m, reflecting the gradual increase in age with depth (Supplementary Table S2).

**3.5 Deep water faunas**

Video footage was taken during all dives using high-resolution cameras fixed to the ROV. At the six dive sites, the ROV both surveyed and collected macrobenthos from the substrate and canyon walls. The ROV paths commenced at the substrate and meandered around the walls until ascending to the top of the canyon. The faunas observed throughout the dives included echinoderms (sea lillies, brittle stars, basket stars, starfish, urchins, benthic and pelagic sea cucumbers), crustaceans (crabs,

lobsters, barnacles), sponges, cnidarians (corals, zoanthids, sea anemones, hydroids, sea pens), molluscs (bivalves, cephalopods, gastropods), brachiopods, polychaete worms, demersal and pelagic fish, and plankton (salps, krill, ctenophores), a selection of which are shown in Figure 14.



Sea pens (Pennatulacea) were occasionally observed on the soft muddy substrates. Otherwise, sessile marine life was more common on the cliff faces but with patchy distribution, and often concentrated together forming diverse communities. Bivalves (*Acesta*), occasional brachiopods, and groups of cup corals (Scleractinia) in particular were found along ledges and overhangs. Various echinoderms, including basket stars (Eyryalina), were often attached to large coral colonies such as *Corallium* (Site

C and E), illustrating the tendency of deep-sea marine life to cluster around habitat forming species. Sponges were also commonly observed around the canyon, with large populations of hexactinellids at Site F and glass sponges (*Walteria*) with commensal shrimp at Site D.

ROV sampling during the dives mainly targeted deep-sea corals, especially those that precipitate carbonate skeletons given their suitability for geochemical proxy studies. Scleractinians (aragonitic) and octocorals (calcitic) were collected from depths

between 674 m and 1815 m from the six dive sites, being mostly sampled from the canyon walls. The most common live scleractinian coral was the cup coral *Desmophyllum dianthus*, which was typically found in clumps near ledges. Other solitary corals were *Caryophyllia* sp.*, Polymyces* sp.*,* and *Vaughanella* sp. Live specimens of the colonial species *Solenosmilia variabilis* were uncommon. Gorgonian (*Corallium, Narella*) and bamboo (*Lepidisis*) corals inhabited most dive sites from deep to intermediate water depths. *Paragorgia, Chrysogorgia*, and proteinaceous black corals were also present. Extensive

fossil deposits comprising both colonial (*S. variabilis*) and solitary (*D. dianthus*) species were discovered in deep (Site C) and intermediate (Site F) waters.

Live corals were collected from intermediate waters (AAIW) at dive sites A, B, D, F, and from deep waters (UCDW) at sites C and E. The carbonate chemistry determined from these waters (Sites B, C, and E) show that the calcite saturation state is above unity ($\Omega_{calcite} > 1$) at all depths sampled (i.e. to 2000 m), whereas aragonite saturation reaches unity ($\Omega_{aragonite} \sim 1$) at

~1000 m (Fig. 11). Thus, it is notable that some aragonitic scleractinian corals (*D. dianthus* and *Vaughanella* sp.) were observed and/or collected from depths (~1360-1770 m) below the aragonite saturation horizon. Similar observations have been made from the Tasman seamounts where corals inhabit waters in which both aragonite and calcite are undersaturated by up to 40% (Thresher et al., 2011).

### 3.5.1 Site-specific habitat surveys and sampling

Near the head of the canyon at Site A, crabs (e.g. *Chaceon albus*) were the common mobile benthos and demersal species were represented by grenadiers and deep-sea dories, and pelagic sea-cucumbers. The canyon walls harboured polychaete tubes worms, glass sponges including a new species *Amphidiscella* sp. nov., occasional echinoderms (urchins, crinoids, and sea stars), lobsters (Fig. 14a), and the zoanthid bearing hermit crab *Sympagurus* sp. (Fig. 14b). The first cnidarians were seen at 716 m (sea anemone and cup coral *Polymyces* sp.), with species richness greater near the top of the cliff where antipatharians

(e.g. Fig. 14c, d) and solitary live and sub-fossil scleractinians (*D. dianthus, Caryophyllia* sp., and *Polymyces* sp.) were common. Soft corals were uncommon and colonial scleractinians were not observed. A selection of cup corals, *D. dianthus, Caryophyllia* sp., and *Polymyces* sp. (e.g. Fig. 14e, f), were collected between 716 m and 674 m, the gorgonian *Narella* sp. was collected at 695 m, a non-branching bamboo coral (*Lepidisis* sp.) was collected at 679 m, and one large black coral



(antipatharian) was sampled at 678 m (Fig. 14d). Two *Acesta* sp. nov bivalves (Fig. 14g) and a hexactinellid sponge were collected between 675-695 m.

At Site B, in slightly deeper waters and along the (east) canyon wall directly opposite site A, were specimens of live an dead scleractinian corals, serpulid tube worms, brachiopod, veruccid barnacle, crustaceans (crabs and squat lobster) and the first

sighting of the large glass sponges (*Walteria*) near the edge of the ridge. The latter (also present at Site D) is the first record of this genus in Australian waters and the Indian Ocean, having previously been reported only from Japan, Kermadec Ridge, and Hawaii. A sea anemone (1079 m) and a gorgonian (probably *Narella* sp. at 1010 m) were the deepest cnidarians observed. Coral diversity and abundance were markedly higher from ~950 to 930 m. Within this range, several live but ~20 dead (i.e. ferromanganese coated) *D. dianthus* were observed, and a dead colony of *S. variabilis* was collected at 936 m. This colony

also contained live and sub-fossil *D. dianthus* with numerous newly settled polyps (seemingly *D. dianthus*), seen as small round dots on dead coral skeletons. Between 930-950 m were a moderately large number of bamboo corals (nominally *Keratoisis*), one collected at 933 m (Fig. 14h), with gold corals (*Chrysogorgia* and *Iridogorgia)* and purple soft coral seen between 937 to 930 m.

At the deepest dive site, in the south at the junction of the two main canyon tributaries, Site C (Dog-Leg Canyon), cnidarians

were observed over most of the depth range. Depth ranges for different and widely distributed taxa included sea anemones (1537-1817 m), live bamboo corals (nominally *Acanella*, *Keratoisis* and *Lepidisis* at 1536-1816 m), live black corals (1572-1789 m) and live *D. dianthus* (1556-1773 m). Over smaller depth ranges, or in smaller numbers, were *Narella* (1749-1766 m), the gold corals *Chrysogorgia* and *Metallogorgia* (1605-1796 m; the former collected at 1793 m), *Anthomastus* spp. (1567 m), sea pens (Pennatulidae, at 1528-1612 m), *Paragorgia* (collected at 1739 m), *Corallium* (1574-1690 m), a large purple soft

coral (1561 m), unidentified soft corals (1548-1606 m), and gorgonians (1568-1577 m). Crinoids and brittle stars commonly colonise dead and live octocorals (Fig. 14i, j), isidids (Fig. 14k), and hexactinellids (Fig. 14l). Most notable was a spectacularly diverse community of organisms hosted by a large *Corallium* octocoral (collected at 1557 m), which included Venus flytrap sea anemones (*Actinoscyphia* sp.), a new species of bivalve mollusc (*Acesta* sp. nov.), basket stars, crinoids, and a new species of hexactinellid sponge (*Farrea* sp. nov., Fig. 14i). This site was where extensive fossil coral deposits were first discovered.

Contained within fine sediment slopes between the canyon walls between ~1560-1790 m, were potentially thousands of mostly cup corals, apparently *D. dianthus* (collected at 1788 m, Fig. 14m), which indicates extensive reef development in the past. Scattered on the cliff face were also a large number of bases of sub-fossil bamboo corals (1550-1795 m), much larger than the live specimens observed.

Nearby on the opposite (southeast) side of the canyon wall but within a shallower depth range, is Site D that was notable for

the stunning display of *Walteria* sp. glass sponges (Fig. 14n), which were especially numerous along the ridge crest. One specimen collected at 986 m was hosting a commensal shrimp, *Paralebbeus* sp. nov. Demersal faunas, such as oreodories and grenadiers, were often observed throughout this dive. Of the cnidarians, the occasional sea anemone (maximum depth 1209 m), black coral (1186 m), and an unidentified "soft coral" (1131 m) were observed. The diversity and abundance of megabenthos increased markedly from ~1080 m depth, remaining high to the top of the cliff face at ~990 m. Within that depth



range were numerous live *Anthomastus* spp. (1004-1072 m), a stylasterid (1022 m), gold coral (*Metallogorgia* spp. at 1077 m), a moderately large number of live bamboo corals (nominally *Keratoisis* and *Lepidisis* at 1046-1077 m), and other octocoral species (probably *Corallium* spp. at 988 m). Several large sub-fossil bamboo corals were observed at the base of the cliff. Both live and dead octocoral samples were collected at 1050 and 1040 m. Some dead scleractinian cup corals, likely *D. dianthus*,

were observed at 1049-1178 m, and seemingly long-dead colonies of the colonial *S. variabilis* occurred at 1040-1073 m. Part of a recently dead colony of *S. variabilis* was collected at 1073 m (Fig. 14o).

Approaching the canyon mouth is the westernmost dive site, Site E, which has steep topography and a deep survey depth range similar to Site C. Sparse megabenthos at ~1600 m were mainly comprised of occasional whip-like black coral. Live cup corals (*D. dianthus* and *Vaughanella* sp.) were sparse (1357-1603 m), and occasional dead specimens were attached to the bedrock

(Fig. 14p). A specimen of *Vaughanella* was collected live at 1472 m, and dead specimens of *D. dianthus* were collected at 1498 and 1444 m. Live bamboo coral (nominally *Keratoisis* and *Lepidisis*) were seen at 1381 m, one collected at 1377 m, and scattered individuals were moderately abundant up to the shallowest survey depth of the dive (1228 m). A *Corallium* collected at 1357 m (Fig. 14q) hosted a small community including colonial and cup corals (*S. variabilis* and *D. dianthus*), a new species of hexactinellid glass sponge (*Farrea* sp. nov.) with its commensal polynoid scale worm, a large ophiuroid (*Ophioplinthaca*),

stalked barnacle (*Glyptelasma orientale*), and squat lobster (*Munidopsis*). A dead manganese-coated octocoral basal stump was collected at 1247 m (Fig. 14r).

Above the canyon rim on the northern plateau, Site F represents one of the shallower dive sites where extensive fossil coral deposits carpet the substrate, in places constituting up to 100% of cover, with large numbers of the live hexactinellid sponge, *Aphrocallistes* sp., comprising the dominant benthos (Fig. 14s). These remnants of extensive ancient reefs are dominated by

*S. variabilis* but also contain large numbers of *D. dianthus* as well as *Caryophyllia*. Large scoops of fossil coral rubble and accompanying sponges were collected from 746-691 m, the associated hash having contained various taxa including gastropod (e.g. the corallivorous epitoniids and *Coralliophila licinus*), bivalve, and pteropod shells. A number of sub-fossil octocoral bases were observed at 692 m (*Keratoisis magnifica*), 690 m (*Corallium*), one thick and highly eroded sample collected at 683 m, but no bamboo corals were observed. Live cnidarians were sparse, with few representatives of antipatharians (742-745 m,

e.g. Fig. 14t), sea anemones (702-751 m), soft corals, unidentified octocorals (682-747 m), and cup corals (*Polymyces* sp., collected at 739 m), and stylasterids were observed throughout the dive. Mobile benthos included a various crustaceans, including squat lobsters, hermit crabs (*Sympagurus*) with the commensal zoanthid (*Epizoanthus*), a solitary king crab (*Paralomis dofleini*) that is the southernmost record of the species, and few occurrences of echinoderms (sea stars and sea urchins). Large amounts of plankton in the water were observed, such as krill, ctenophores, and salp chains. Pelagic faunas

observed included holothurians (Fig. 14t), various fish (e.g. Bigeye Ocean Perch, *Helicolenus* sp., Fig. 14u), and a shark.

Further details of observations and samples collected during the Perth Canyon cruise (FK150301) can be found in the events logs and final cruise report, which can be accessed from the Schmidt Ocean Institute website. Taxonomic analysis of the collected specimens is ongoing, with species descriptions for the hexactinellid sponges, *Acesta* sp. nov., and crinoids currently in preparation.



### 3.5.2 Ages of fossil deposits

U/Th ages determined from the skeletons of fossil colonial (*Solenosmilia variabilis*) and solitary (*Desmophyllum dianthus*) scleractinians from the two coral graveyard deposits (Sites C and F) are similar, although specimens from the deeper waters span a wider age range (Fig. 15; Supplement Table S3). Corals collected from deposits of mostly *S. variabilis* (Fig. 14s), which

covered the substrate of the plateau north of the canyon (Site F) at intermediate water depths (690-720 m), range in age from ~22 ka to ~27 ka (BP). The *D. dianthus* (Fig. 14m) dominated deposits, which protruded from sediment slopes between the canyon walls in deep waters (1560-1790 m) within the southern part of the canyon (Site C), yielded ages from ~18 ka to ~33 ka (BP).

The abundance of these deposits through a large depth range indicates that both colonial and solitary cold-water corals

flourished during the last ice age, spanning the Marine Isotope Stage (MIS) 3-2 transition and the Last Glacial Maximum (LGM) into early Heinrich Stadial 1 (HS-1), with the latter marking the onset of the deglaciation. These ages are equivalent to those of *D. dianthus* recovered from high latitude waters of the Southern Ocean south of Tasmania (Hines et al., 2015) that were collected from depths (~1400 to ~1800 m) equivalent to the deeper *D. dianthus* deposits (Site C) of the Perth Canyon.

## 4 Conclusions

The Perth Canyon is a large, submarine, and remarkably quiescent 'fossil' canyon system located on a passive margin that partly incises the continental shelf. It hosts sparsely populated yet relatively diverse communities of deep-sea biota, typically concentrated in 'hot spots' along the canyon walls and rim, as well as hard substrates of the surrounding continental slope. The most spectacular occurrences featured large specimens of the scleractinian solitary coral *Desmophyllum dianthus*, rows of tall *Walteria* glass sponges along the canyon rim, diverse communities including echinoderms and anemones hosted by corals

(*Corallium*), as well as hexactinellid sponges that carpeted the substrate of the canyon plateau. The fine soft muds of the canyon floor, however, were often largely barren of epifaunal megabenthos, with isolated occurrences of mostly sea pens, crustaceans, echinoderms, and other octocorals. Calcifying megabenthos, such as scleractinians and octocorals (e.g. bamboo and *Corallium*), occur throughout the intermediate and deep waters, with some scleractinians living below the aragonite saturation horizon.

Extensive coral graveyards discovered at two widely separated sites and depths indicate that deep-water corals thrived in the canyon during the last ice age (~33 to ~18 ka BP) thus prior to and during the Last Glacial Maximum. The substrate of the canyon plateau at the northern shallower (~700 m) site is blanketed mostly by colonial *Solenosmilia variabilis* with subordinate cup corals (*D. dianthus*). Conversely, the deep fossil deposits (~1700 m) within the canyon are dominated by *D. dianthus*, which protrude from fine sediments draping the canyon walls and ledges indicating a quiescent environment at these depths.

The carbon isotope ($\delta^{13}C$ and $\Delta^{14}C$) compositions of the canyon waters reveal that anthropogenic carbon has entered upper intermediate water depths. Changes in the conditions and chemistry (temperature, circulation patterns, and carbonate saturation state) of polar sourced intermediate and deep waters induced by climate change and $CO_2$ uptake have important consequences




for the deep-sea calcifiers inhabiting the canyon given the already adverse seawater conditions ($\Omega \approx 1$) under which they must form their carbonate skeletons. Ongoing geochemical studies of both young and fossil corals collected from the Perth Canyon are providing important environmental records of intermediate to deep waters through major climate events, from the last glacial period as well as pre-industrial to industrial intervals. Such data are needed to help discriminate long-term natural

variations from recent anthropogenic effects that are beginning to impact the deep oceans. Deep-water corals can thus play an important role both as archives of environmental change in these poorly studied yet critical areas within the ocean-climate system, and in determining the inherent vulnerabilities and future responses of deep-water ecosystems to global climate change.

**Data availability.** The imagery and seawater data collected during the R/V Falkor cruise (FK20150301) and cruise reports
can be found on the Schmidt Oceans Institute website: https://schmidtocean.org/cruise/perth-canyon-first-deep-exploration/, http://www.marine-geo.org/tools/search/entry.php?id=FK150301, seawater metadata, bottle, and CTD-sensor data also at: http://catalogue.aodn.org.au/geonetwork/srv/eng/metadata.show?uuid=579fc6fe-3f22-4101-94d1-3dc85b7d0b36 (metadata), http://thredds.aodn.org.au/thredds/catalog/UWA/RV_Falkor/2015_03_Perth_Canyon/Bottle_data/cataloghtml (bottle data), and http://www.rvdata.us/catalog/FK150301 (CTD data), and subsequent seawater and carbonate chemical analyses are given
in the Supplementary file. Additional seawater data collected from earlier cruises are from the GEOSECSv2 and GLODAPv2 publically available data repositories. Faunas, rocks, and sediments collected during the cruise have been archived by the Western Australian Museum (WAM) and the University of Western Australia (UWA). Samples collected specifically for geochemical analysis reside with the cruise Chief Scientist, Professor Malcolm McCulloch, at UWA (malcolm.mcculloch@uwa.eu.au).

**Competing interests.** The authors declare that they have no conflict of interest.

**Acknowledgements.** The authors gratefully acknowledge the Schmidt Ocean Institute for providing the R/V Falkor and all necessary equipment and crew that enabled us to undertake this cruise (FK150301). The Falkor crew and the ROV pilots from
Neptune Marine Services are thanked for their expert assistance during the cruise. The Australian Research Council is acknowledged for fellowship funding to M. McCulloch (FL120100049) and J. Trotter (FT160100259). We also acknowledge the Australian Institute of Nuclear Science and Engineering for AINSE Research Award 16/009 (M. McCulloch, J .Trotter, J. Falter, R. Thresher, M. Taviani, P. Montagna) to undertake [14]C analyses of seawater. Kai Rankenburg (UWA) is thanked for assistance with U/Th mass spectrometry. Ana Hara, Jane Fromont, Oliver Gomez, Lisa Kirkendale, Glenn Moore, and Corey
Whisson (from WAM) are thanked for assistance with identifications and specimen processing. This is ISMAR-CNR, Bologna scientific contribution n. #.





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

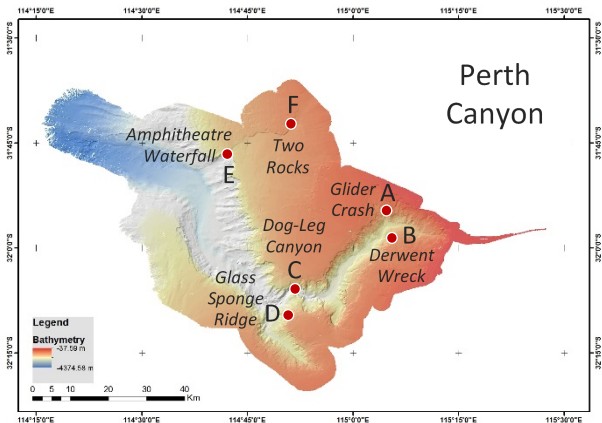

**Figure 2: Multi-beam map of canyon showing ROV faunal collection sites: Site A (dives 2 and 8), B (dive 4), C (dive 6), D (dive 7), E (dive 5), and F (dive 9).**





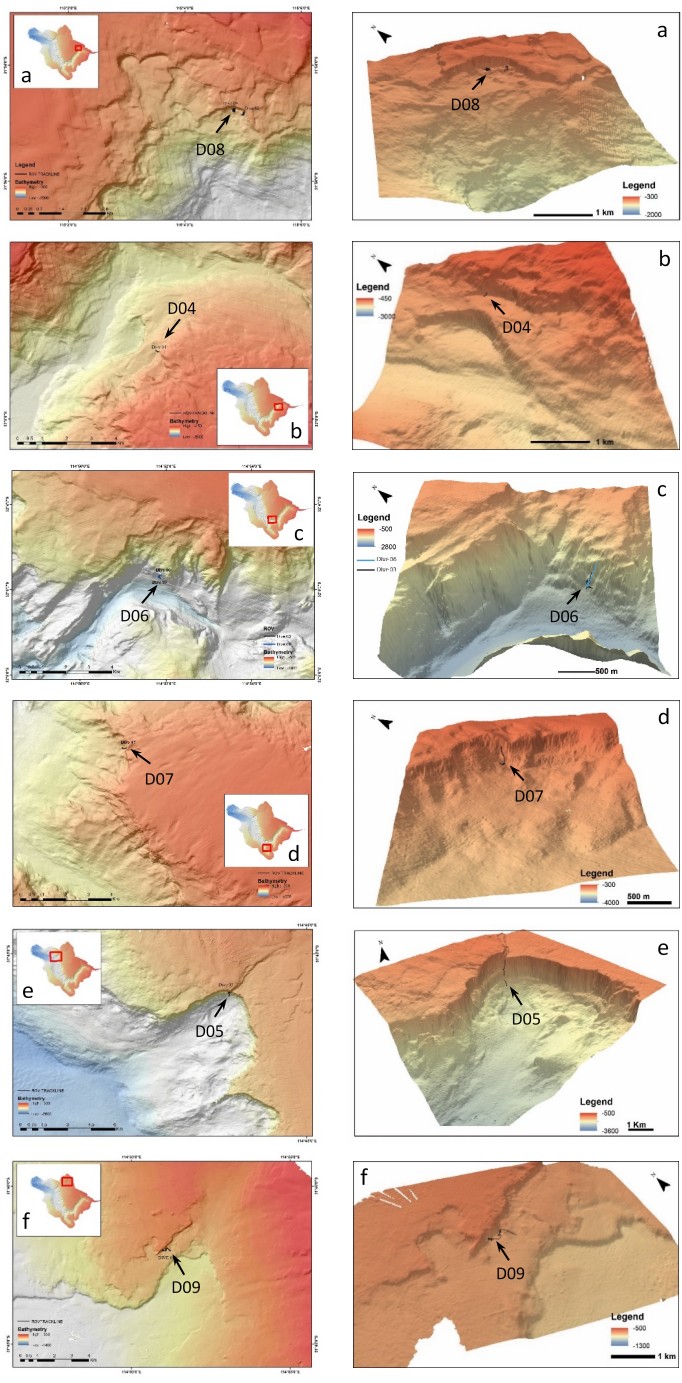

**Figure 3: High-resolution multi-beam maps of canyon at the six ROV sample collection sites (A – F). Black line near arrow depicts ROV track, with each dive number indicated.**





**Figure 4: Outcrops and lithological features of the canyon walls and substrate at ROV sample collection sites. A-C: Site A (Glider Crash); D-F: Site B (Derwent Wreck); G-I: Site C (Dog-Leg Canyon); J-L: Site D (Sponge Ridge); M-O: Site E (Amphitheatre Waterfall); P-R: Site F (Two Rocks).**



**Figure 5: Regional stratigraphy and age ranges determined from geological samples from the Perth Canyon reported in prior studies (modified from Shafik, 1991) and from the present study. Dashed line illustrates range of biostratigraphic ages from Marshall et al., 1989. Open circles denote approximate ages determined from foraminifers, solid circles and bars indicate Sr isotope age ranges of samples collected during the R/V Falkor cruise.**





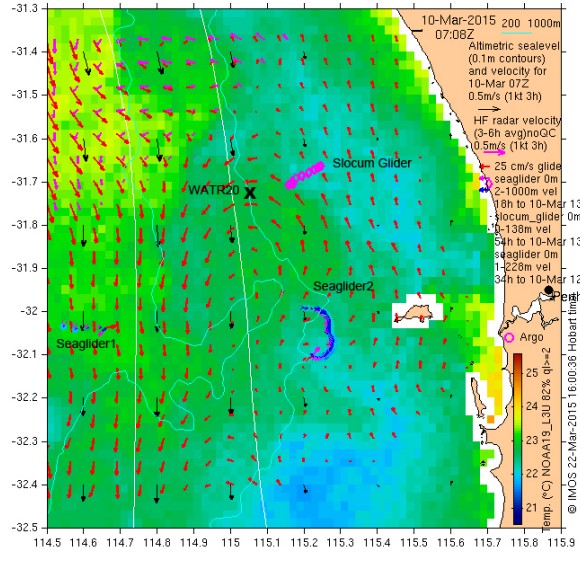

**Figure 6: Surface currents measured by HF Radar systems overlain on satellite derived sea surface temperature map of the study region obtained on 2 March 2015. The colder water represents the northward flowing Capes Current whilst the warm water represent the Leeuwin Current. An eddy is located in the vicinity of the Perth Canyon. Small white squares represent the location of moorings (current meters and thermistor chains).**

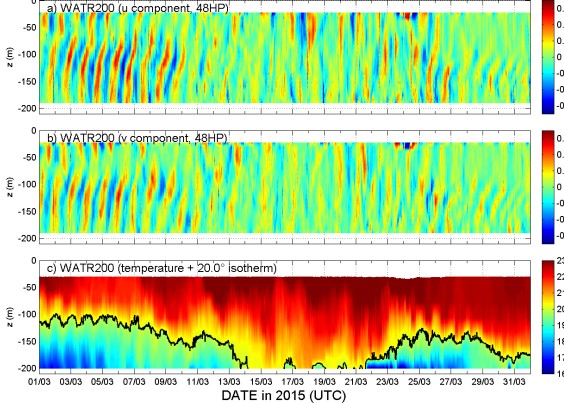

**Figure 7: Time series of low passed currents (a,b) and temperature (c) at the WATR20 station during March 2015 showing a period of strong diurnal currents through the water column and associated upwelling during the period of the voyage (1-12 March).**

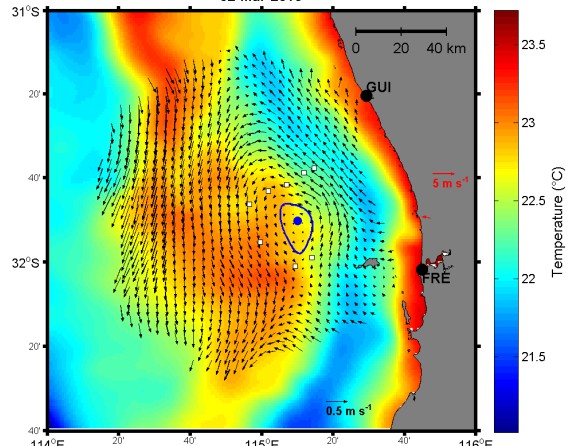

**Figure 8: Satellite derived sea surface temperature map of the study region obtained on 10 March 2015 towards the end of the voyage. The red/magenta arrows indicate surface currents as measured by HF Radar. Locations of 3 ocean gliders (1 Slocum and 2 Seagliders) are shown. The WATR20 is the location of an oceanographic mooring along the 200m depth contour from which data shown on Figure X is derived.**

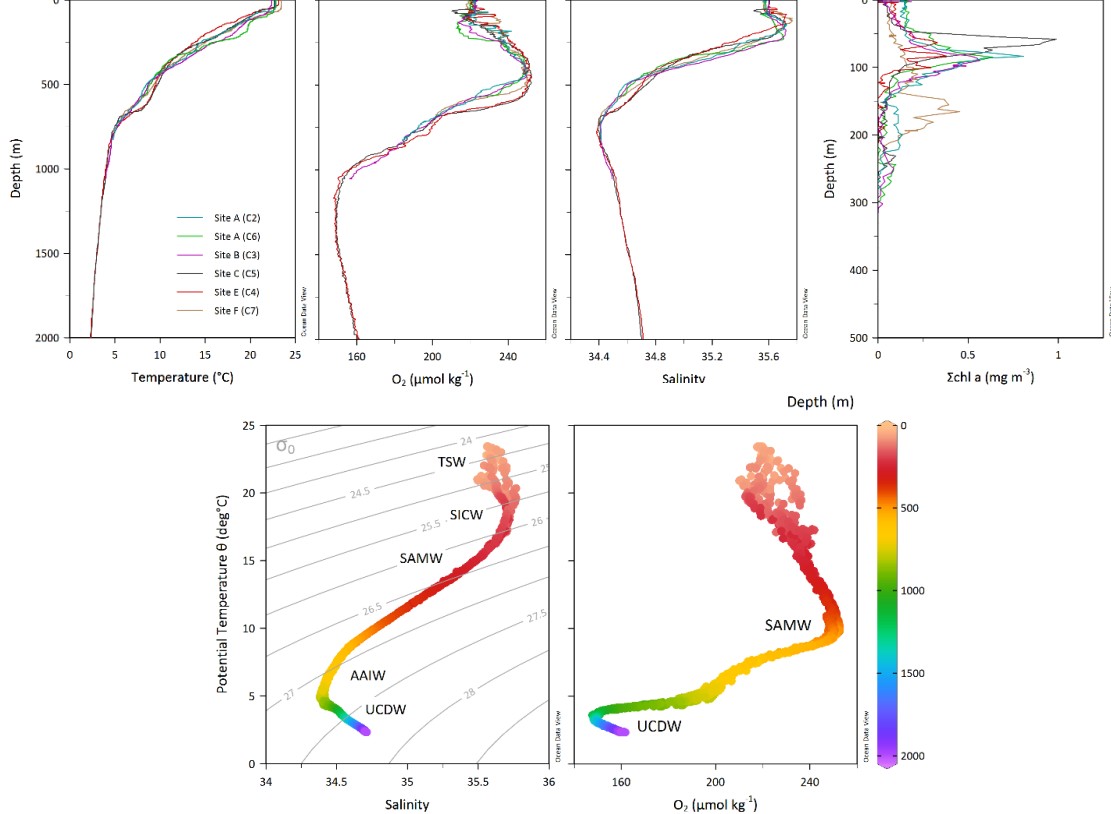

**Figure 9: Top: Vertical profiles of temperature, salinity, dissolved oxygen (O₂), and total chlorophyll a (Σ_chla), recorded by the CTD at five ROV dive sites. Note the shallower vertical scale (depth) on the plot of chlorophyll a. Bottom: temperature–salinity and temperature–oxygen plots for Potential Temperature Depth, showing water masses identified in the Perth Canyon. TSW: Tropical Surface Water, SICW: Subtropical Indian Central Water, SAMW: Subantarctic Mode Water, AAIW: Antarctic Intermediate Water, and UCDW: Upper Circumpolar Deep Water. Isopycnals calculated with the reference pressure at 0 m.**

**Figure 10: Left: Dissolved nitrate plus nitrite ( $NO_x^-$ ) and soluble reactive phosphorus (SRP) vs. depth at five ROV dive sites in the Perth Canyon. 95% of all nutrient samples had ammonium concentrations less than 0.5 μM.**

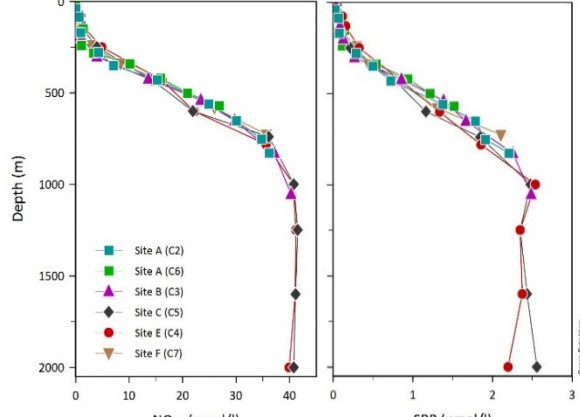



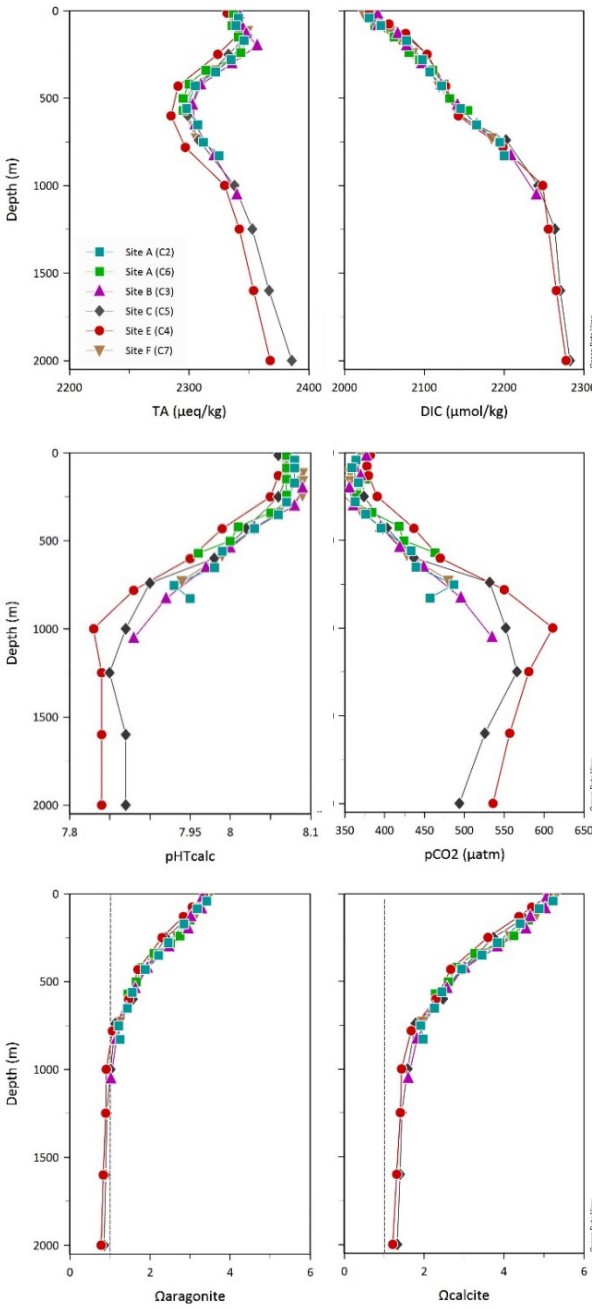

**Figure 11: Vertical profiles of seawater compositions for total alkalinity (TA), dissolved inorganic carbon (DIC), calculated pH on the Total Scale (pHT$_{calc}$), partial pressure of carbon dioxide (pCO$_2$), and carbonate saturation state for aragonite ($\Omega_{arag}$) and calcite ($\Omega_{calc}$) measured at five ROV dive sites.**



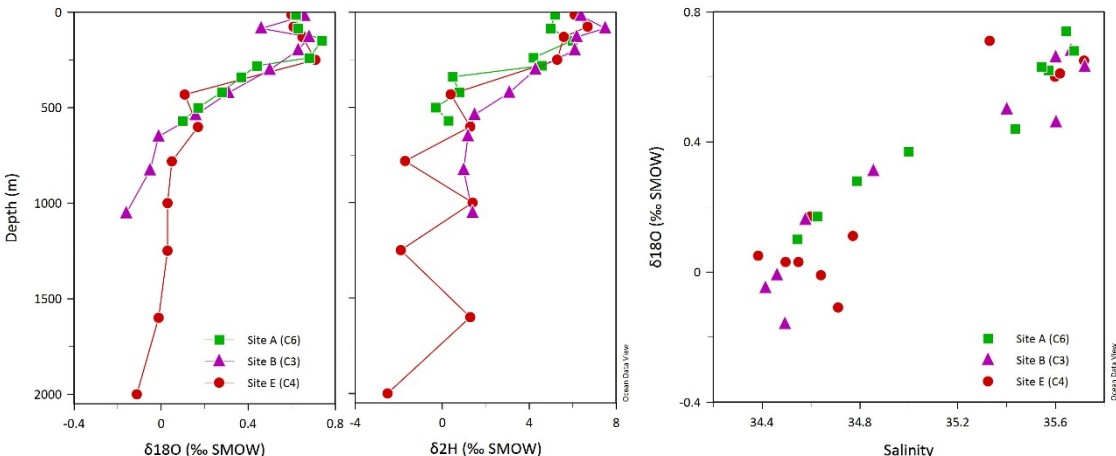

**Figure 12: Left: Stable isotope composition of oxygen (mainly water, δ¹⁸O) and hydrogen (δ²H) in seawater at three of the ROV dive sites (A: Glider Crash, B: Derwent Wreck, E: Amphitheatre Waterfall). Right: Seawater δ¹⁸O isotope versus salinity plot.**

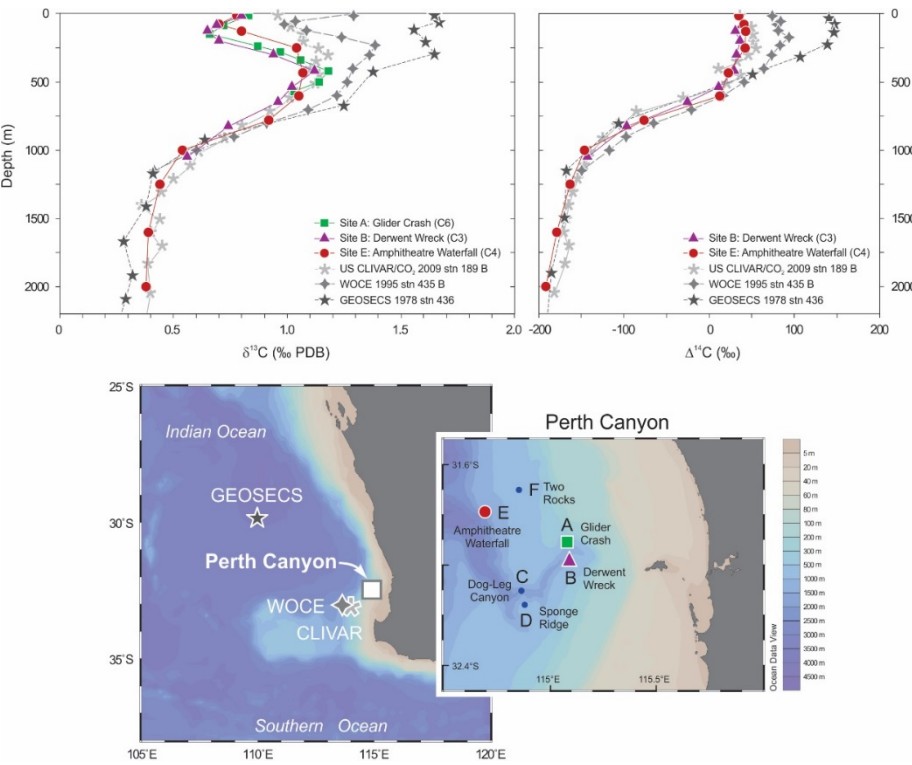

**Figure 13: Top: Representative vertical profiles of seawater δ¹³C (left) and Δ¹⁴C compositions (right) from the Perth canyon compared with data from earlier cruises in the region: GEOSECS-78 (1978, station 436), WOCE-95 (1995, station 435), and US CLIVAR/CO2 (2009, station 189). Satellite map show relevant sampling sites from the Falkor (Perth Canyon), GEOSECS, WOCE, and CLIVAR cruises.**







**Figure 14. Images of faunas at the six ROV dive sites. Lobster *Projasus parkeri* (a); hermit crab *Sympagurus* sp. with zoanthid *Epizoanthus* sp. (b); antipatharian corals (c, d); cup corals *Desmophyllum dianthus* (e) and *Polymyces* sp. (f); Acesta sp. bivalve (g); bamboo coral (h); *Corallium* 'community' hosting various taxa (i); crinoid on *Corallium* sp. (j); crinoids and brittle stars on dead bamboo coral (k); hexactinellid sponge (l); fossil *D. dianthus* deposits (m); *Walteria* sp. glass sponge with comensal shrimp and bamboo coral (n); *Solenosmilia variabilis* colony (o); brittle stars, *Anthomastus* sp. and dead cup corals (p); *Corallium* sp. with brittle stars and *D. dianthus* (q); Mn-coated dead *Corallium* stump (r); hexactinellid sponges, crinoids, and fossil *S. variabilis* deposits (s); pelagic holothurian and antipatharian coral (t); Bigeye Ocean Perch *Helicolenus* sp. and stylasterids (u).**





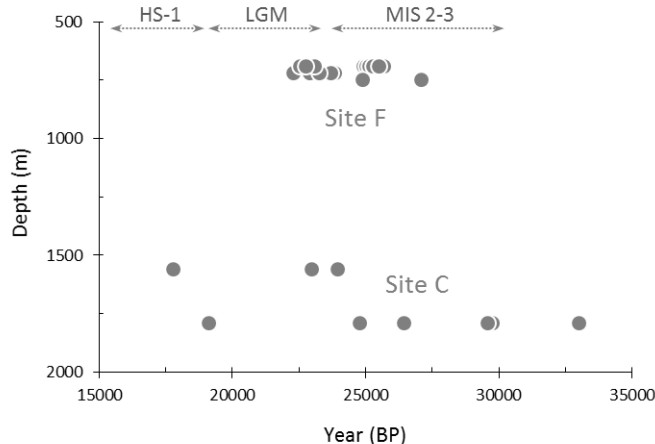

**Figure 15. U/Th age relative to depth distribution of scleractinian corals from the Perth Canyon collected from all dive sites. Fossil (MIS 3 to HS-1) reef deposits occur at sites C (~1500-1800 m) and F (~700 m).**