# Peer review of "Unveiling the Perth Canyon and its deep-water faunas"

_Biogeosciences, 2018_

## Referee Comment (RC1) · Anonymous Referee #1 · 15 Sep 2018

This is an almost very complete analysis of the Perth Canyon system, which describes accurately the chemical characteristics of the different water masses, biostratigraphic and palaeoecological characteristics of the system and the biological communities inhabiting this dynamic habitat. Additionally, it provides an accurate comparative analysis of the changing carbonate chemistry of the deep-sea habitats that occupy this canyon system. However, what is the role of the canyon on carbon cycling in the context of modern global climate change has been poorly discussed as well as other sections (i.e. section 3.4) that I believe that it would need to be more discussed in light of present studies on the topic. Also, I expected to read more on how the geophysical condition drives the biological systems. For example, I suggest attempt in the discussion to linking geological structures and water mass with community structure and species

distribution (i.e. Baker et al. 2011, Robert et al. 2014, Brooke and Roos 2014). I also recommend reorganizing and shortened some parts of the manuscript (see my comments below), as well as improve some figures. In light of these comments, I suggest the manuscript be accepted after a moderate revision process.

Please find below several more specific suggestions: Pag. 4 Line 20-25. These are details of sampling methodology. I recommend moving this part down and slightly modified it. Pag 9. Line 9-13. Move to the Material and Method section and slightly modified it Pag 9. Lines 14-24 I think you need to move this text down and integrate into next section where the geological description of the dives is described. It is redundant in some parts. Pag13. Line 18-25. Please include the number of the figure or references that support this statement. Additionally, it would be interesting adding a brief discussion about how the eddies affect the productivity within the canyon (has have been observed in other canyons i.e. Amaro et al., 2017) Pag 15. Section 3.4.3. Could you explain the effect of the canyon on the carbon cycle a little more? There is still an important knowledge gap related to this topic on canyon ecology. Pag. 16-17. Line 26 al 16. Don't think most of this is needed, there is some information repeated below. I recommend delete some parts and move down the information you think is relevant and integrate into next section. Pag 17-18. Section3.5.1 I think that the structure of this section can improve and better describe the relation between topographic/hydrographic characteristic of the site and the community/species founded. For this purpose, I really recommend to modified Figure 14 including each transect and the photo with the main characteristics species or communities inhabiting the different geological structures within transects. Figure 2. I suggest deleting Figure 2 (not relevant data). Also, add the sampling point in Figure 1B. Additionally, I believe that the quality of Figure 1 can be improved. If possible, try to follow similar format within the Figures (i.e. regarding the position of the legend, the names of the categories, etc). Increase the size font of lat/long (not legible). I suggest including also north in the map. Figure 3 and 4. I would merge Fig 3 and 4, makes it easy for readers to quickly understand the dives characteristics (e. g. Figure map and 2 photos of each dive) Supplement Figures S1

and S2. It almost repeated from Figure 4. I would delete it. No new information is provided.

Below are some minor comments that deserve attention. Pag 1. Line 21. Changes CTD profiling for chemical Pag 3. Line. 1 Don't think sub-heading is necessary for Introduction. Pag5. Line 29. 10 subsamples from each site, is correct?. Please clarify. Pag 6. Line 1. Changes to "Vienna Standard Mean Ocean Water (VSMOW)" Pag 7. Line 29. 2.5 Add Stronium (Sr) isotope dating Pag 8. Line 8. Changes Stronium for Sr Pag.14 Line 31: Include; deeper depths Pag.14. Line 28. Delete "s" from FIGs Pag15-Line 1: Changes the sentences to "formation, suggesting that this material is most likely pelagic in origin. Figure legend 14. Acesta Italicize Pg 17. Line 25. Changes: "mobile megabenthos. Demersal.." Line 27 Changes Sea urchins Pag.16. Line 25. Changes; deep-sea fauna Figure 6. - I did not found the Small white squares representing the location of moorings in the Figure. Please include them. Please also include the north in the figure. Figure 7. Increase the size of the letters of the Figures and axis font. Figure 8. Include the north. Include in the legend; Small white squares represent the location of moorings (current meters and thermistor chains). Figure 9. The legend and colors of Figure 9A is not legible, increase the size Figure 13. Rename it appears as Fig 12.

---

## Referee Comment (RC2) · Anonymous Referee #2 · 25 Oct 2018

General comment: The ms presented an excellent, one-of-the-kind data set including numerous ROV dives at 5 different sites and depths along the Perth canyon, and ship-board CTD casts (with water samples) collected from those sites that allowed a variety of physical, chemical, and isotope analyses of the entire water column. It is a rather comprehensive field report of very good quality, but unfortunately somewhat short to be considered a "research article" that this ms is intended to be. After double checking the scope of the journal and its requirement for a research article manuscript ("...on all aspects of the interactions between the biological, chemical, and physical processes ... to cut across the boundaries of established sciences and achieve an interdisciplinary view of these interactions"), I reluctantly had no choice but rejecting the current form of the ms.

[Figure]

Specific comments: 1. I would recommend the author to think about, in their revising ms, what's the story in it? Namely what's the hypothesis or science questions the ms wants to address? What's the importance or relevance in solving those hypotheses or addressing the science questions. 2. The ms had in-depth descriptions of the CTD/water sample works with respect to the physical, chemical, and isotope analyses. Except for the isotope discussion related to LGM (very nicely done!), there was hardly discussions linking those properties of different disciplinarians. 3. One key difference between research articles and data reports is concise vs overdone details. There are just too many details in the ms that should go to a supplemental file.

---

## Author Comment (AC1) · 25 Nov 2018

Reviewer's comment: The ms presented an excellent, one-of-the-kind data set including numerous ROV dives at 5 different sites and depths along the Perth canyon, and shipboard CTD casts (with water samples) collected from those sites that allowed a variety of physical, chemical, and isotope analyses of the entire water column. It is a rather comprehensive field report of very good quality, but unfortunately somewhat short to be considered a "research article" that this ms is intended to be. After double checking the scope of the journal and its requirement for a research article manuscript ("...on all aspects of the interactions between the biological, chemical, and physical processes ... to cut across the boundaries of established sciences and achieve an interdisciplinary view of these interactions"), I reluctantly had no choice but rejecting the

current form of the ms.

Authors' response: This study is highly cross-disciplinary, consistent with the scope of Biogeosciences articles. There are various accounts of prior publications in this journal being significantly descriptive (and less diverse) where unique datasets are being reported for the first time. Given these are the first of such datasets of a previously unexplored canyon, it poses some constraints on the scope of the paper, as well as the extent we can reliably discuss the implications of, for example, brief snapshots of faunal discoveries from this large canyon system. We therefore feel that the reviewer is somewhat conservative in their interpretation of the journal scope. Nevertheless, we will modify the text to provide a more nuanced cross-disciplinary discussion of our observations, albeit qualified given the constraints of our study.

Manuscript changes: We will more clearly articulate the goals of this study and its significance, and revise the manuscript to enhance the discussion of cross-disciplinary relationships from our observations, while transferring some descriptive details to the supplementary file as requested. We believe that our revisions to the manuscript will address this reviewer's concerns, as detailed below.

Specific comments:

Reviewer's comment: I would recommend the author to think about, in their revising ms, what's the story in it? Namely what's the hypothesis or science questions the ms wants to address? What's the importance or relevance in solving those hypotheses or addressing the science questions.

Authors' response: We appreciate that the aims and significance of the study could be more explicitly described, and particularly more broadly in the context of canyon research undertaken elsewhere. The key aims of our study were primarily to: (i) explore the inhabitants of this canyon system, and whether cold water corals in particular are present. In this sense the study was literally a voyage of discovery being the first deep-water exploration of this canyon; and (ii) whether we can detect anthropogenic effects

in these waters and discuss their implications. This required characterising the physical and chemical properties of seawaters, which also provide important environmental context for those inhabitants observed and collected as well as essential baseline information for subsequent studies. The intent of this paper is thus to provide an overview of the canyon environments and the inhabitants discovered, and to discuss any potential anthropogenic effects we measured.

Manuscript changes: Refocus the text to more explicitly state and address the key science questions that this study presents and can answer, and the significance of those outcomes (as above). This includes expansion of the discussion on anthropogenic impacts, such as the implications of $\delta$13C and $\Delta$14C data regarding anthropogenic carbon and the Suess effect. Additional text will also contrast the Perth Canyon with other canyon systems, to provide some context within broader submarine canyon research. We will also revise the conclusions to more succinctly summarise the outcomes and scientific significance of this study.

Reviewer's comment: The ms had in-depth descriptions of the CTD/water sample works with respect to the physical, chemical, and isotope analyses. Except for the isotope discussion related to LGM (very nicely done!), there was hardly discussions linking those properties of different disciplinarians.

Authors' response: Characterising the seawater compositions is important for understanding the environmental context of the canyon and its inhabitants, and for addressing the science question regarding the potential extent of anthropogenic impacts on the canyon waters, the latter having been discussed in section 3.4.3 and 3.5.

Manuscript changes: We will further clarify and expand our discussion of various seawater parameters, such as water masses, temperature, pH, saturation state, both in the context of seawater hydrography as well as the fauna observed at each dive site (and hence faunal depth distributions), and their broader implications. These revisions would better link our observations and measurements across the various disciplines.

Reviewer's comment: One key difference between research articles and data reports is concise vs overdone details. There are just too many details in the ms that should go to a supplemental file.

Authors' response: We do recognise this issue raised. We are faced with the dilemma that the data sets from this study, being the first exploration of this canyon system, are not published in the scientific domain but need to be fully reported to support the outcomes and broader discussions of this and subsequent papers. We therefore believe that it is important to comprehensively present the data here, which will be an important reference for other researchers as well as underpin subsequent specialist papers from our group on this poorly studied region.

Manuscript changes: We will transfer details described in the methods, geology, and seawater descriptions, into the Supplementary file as suggested by the reviewer, and thus provide only a synopsis of the salient information for those sections in the main text. The main discussion would therefore focus on the canyon faunas and their environmental context together with inferences on the anthropogenic carbon detected from our seawater analyses.

---

## Author Comment (AC2) · 25 Nov 2018

Reviewer's comment: ". . .the role of the canyon on carbon cycling in the context of modern global climate change has been poorly discussed as well as other sections (i.e. section 3.4) that I believe that it would need to be more discussed"

Authors' response: Our study depicts only a brief snapshot of canyon conditions at a single point in time (over a 10 day cruise period). The canyon is located in an oligotrophic environment with very low primary production. The absence of river run-off means that the canyon does not receive any terrigenous inputs. The region has a very low tidal range with negligible tidal currents. Thus, the Perth canyon is quite unique in comparison to similar canyons globally. The field measurements indicated very consistent seawater chemistry profiles across the different locations around the canyon. These profiles were also very similar to those that have been recorded in previous studies (Woo and Pattiaratchi, 2008; Rennie et al., 2008). The exception to this pertains to chlorophyll a profiles where variability is detected within the surface waters (to ∼200m depth) above the canyon; the upper water masses however are controlled by poleward flowing surface currents (the Leeuwin Current) that do not directly impact the hydrodynamics of the canyon. The hydrodynamics within these waters, determined mostly from oceanographic modelling, is covered in the literature (e.g. Rennie et al., 2008) and have suggested that eddy formation and intermittent upwelling/downwelling occurs near the head of the canyon mainly driven by the interaction between the Leeuwin Current and continental shelf processes. Such processes therefore have little effect on waters within most of the canyon system (i.e. at deep and intermediate depths, being the focus of this paper), which has a deep canyon rim. The main influence within the canyon is through the northward flowing Leeuwin Undercurrent (300-800m) that transports sub-Antarctic Mode water, SAMW) containing high dissolved oxygen northwards and the Antarctic Intermediate water (AAIW). The Perth Canyon therefore seems to be a predominantly passive system, which importantly lacks input of coastal sediments and organic matter, and only the surface currents interact with the surrounding shelf near the head of the system but have little interaction with the canyon itself. Accordingly, the canyon per se appears to have little influence on carbon cycling, its predominantly intermediate and deep water compositions being inherited from distant sources, their nutrients being almost entirely advected from the south together with minimal local contributions via biological export from the surface waters. Section 3.4.3 specifically discusses the carbon isotope compositions of the waters in the context of global climate change, these compositions being very consistent across the 3 representative (shallow and deep) dive sites, which are also consistent with the global trend in other basins reflecting the lack of local influences.

Manuscript changes: We propose to clarify further the canyon hydrography (as discussed above) in section 3.4 to emphasize that dynamic interactions are mostly restricted to the upper water masses and near the head of the canyon, and highlight the lack of sediment and nutrient input from coastal sources which contrasts to many other canyon systems described in the literature. The discussion of the seawater carbon isotope ($\delta$13C and $\Delta$14C) compositions and their implications for global climate change will be expanded, and in particular the relationship between changes in $\delta$13C and $\Delta$14C (i.e. Suess effect due to emissions of fossil fuel carbon) as well as the sensitivity of $\delta$13C to biological modification. Here we emphasise that most of these changes are still within the upper water masses, and no deeper than the uppermost intermediate waters (< 800 m).

Reviewer's comment: "...more on how the geophysical condition drives the biological systems. For example, I suggest attempt in the discussion to linking geological structures and water mass with community structure and species distribution"

Authors' response: It should be recognised that our key goals were (1) to search for cold-water corals in particular, and (2) characterise the canyon environment together with assessing potential anthropogenic changes that might be registered by the canyon waters. The outcomes would thus serve to provide a baseline for future studies, including specialist geochemical proxy research undertaken on the coral fauna to understand environmental changes occurring in this region. A comprehensive taxonomic and ecological study is therefore not the focus of this paper, whereas we do discuss the initial findings of this first exploration and characterisation of the canyon's environments and inhabitants, including the success of locating and collecting cold-water corals. We also emphasize that this is the first oceanographic cruise and exploration of this canyon system, which offers considerably limited spatial and temporal data compared to other canyon systems described in the literature. Further discussion, specifically on faunal distributions and ecology, is therefore premature and well beyond the scope of this paper. These evaluations would be a major endeavour requiring a separate, stand-alone study producing habitat maps with a comprehensive assessment of all footage by specialist taxonomists. Given that the ROV footage from this cruise is the first visual

representation of the canyon, we could not consider that these brief snap-shots surveyed at one point in time from these spatially limited sites within this very large canyon system would necessarily be representative of the canyon habitats and ecology. The relatively limited data from this first (and short) cruise therefore stand in contrast to that of other canyon systems described in the literature, which have benefited from repeat and more extensive studies over extended periods (e.g. as those cited by the reviewer). The depth range of faunal sampling and their associated water masses, including relevant chemical and physical characteristics, have been discussed. The faunal sampling is primarily within deep and intermediate water depths ($\sim$650 to $\sim$1800 m), and are the main focus of this study. These waters are sourced from the Southern Ocean (SAMW, AAIW and UCDW) and are not affected by local canyon topography as shown by their consistent characteristics at respective depths across the dive sites. Furthermore, changes in local hydrography (eddies, upwelling) tend to be limited to shallower depths above the canyon (limited to depths < 300 m) and often develop further offshore, thereby affecting surface water biota; such processes have in any case been covered in the literature and are cited in this paper. Thus, the deeper waters within the canyon appear to be quiescent and lack strong currents (tides have negligible influence), as revealed by expanses of sediment drapes and our observations during ROV deployments which did not encounter strong turbulence effects below the surface currents (> 300m).

Manuscript changes: We propose to include further details to emphasise the specific points outlined in our response above. We would also reorganise section 3.5 and provide more details of the substrates associated with the fauna and the water masses that they inhabit, with some reference to canyon dynamics near the canyon head and the fauna inhabiting the upper intermediate waters in that area. We propose to add a new figure that shows the key taxa observed at each dive site. We cannot draw reliable inferences beyond those comments.

Specific suggestions:

Reviewer's comment: Pag. 4 Line 20-25. These are details of sampling methodology. I recommend moving this part down and slightly modified it.

Authors' response: Agree

Manuscript changes: Change as requested

Reviewer's comment: Pag 9. Line 9-13. Move to the Material and Method section and slightly modified it

Authors' response: Agree

Manuscript changes: Change as requested

Reviewer's comment: Pag 9. Lines 14-24 I think you need to move this text down and integrate into next section where the geological description of the dives is described. It is redundant in some parts.

Authors' response: Suggestion accepted

Manuscript changes: Rationalise the text for this section (bathymetry) and integrate with the section below (geology), thus removing subsection Section 3.1.1. Integrating these subsections of text is necessary to be consistent with the new figure that combines figures 3 and 4, which is requested by the reviewer (see below).

Reviewer's comment: Pag13. Line 18-25. Please include the number of the figure or references that support this statement. Additionally, it would be interesting adding a brief discussion about how the eddies affect the productivity within the canyon (has have been observed in other canyons i.e. Amaro et al., 2017)

Authors' response: Agree to include citations to support the text. Some inferences regarding the relationship of eddies/hydrography and productivity have been made in one of the few published oceanographic studies of the Perth Canyon. However, such processes are limited to the upper/surface waters above the canyon, hence not within intermediate and deep waters which are the focus of our study. We also emphasise

that such discussions on the Perth Canyon are limited by the lack of data given the very few studies, unlike the extensive database on the Whittard Canyon that the Amaro et al 2016 (2017 not found) paper has synthesised from "a wide range of specific studies over the past 10-15 years, covering many aspects of submarine canyon research."

Manuscript changes: The relevant figure number and citations can be provided to support the text in lines 18-25 (p. 13). Inferences regarding our very limited understanding of the links between surface hydrodynamics and productivity within these upper water masses can be included, as discussed in prior oceanographic studies of the Perth Canyon (Rennie et al., 2008, 2009).

Reviewer's comment: Pag 15. Section 3.4.3. Could you explain the effect of the canyon on the carbon cycle a little more? There is still an important knowledge gap related to this topic on canyon ecology.

Authors' response: Overall, the canyon per se has little effect on carbon cycling as changes in hydrography occur mostly within the upper/surface waters so do not tend to effect much of the intermediate deep-waters habitats which are the focus of this study. This mostly 'passive' system at those depths is also reflected by the consistent seawater compositions between the dives sites as discussed in the paper. The nutrients in the canyon are almost entirely advected from the high latitudes of the Southern Ocean during the formation of intermediate and deep waters, as well as local biological export from the surface waters above the canyon. Also please see reply above (General Query).

Manuscript changes: We will clarify and expand on the details given above, including where possible and appropriate, the potential influence of hydrodynamics and nutrient cycling within the uppermost intermediate water depths where they might affect inhabitants near the canyon head. We can further discuss the implications of the $\delta13C$ and $\Delta14C$ data, regarding anthropogenic carbon and the Suess effect. Also please see reply above (General Query).

Reviewer's comment: Pag. 16-17. Line 26 al 16. Don't think most of this is needed, there is some information repeated below. I recommend delete some parts and move down the information you think is relevant and integrate into next section.

Authors' response: Agree

Manuscript changes: We propose to remove subsection 3.5.1 and hence integrate the text in this section (3.5), which would eliminate any repetition.

Reviewer's comment: Pag 17-18. Section3.5.1 I think that the structure of this section can improve and better describe the relation between topographic/hydrographic characteristic of the site and the community/species founded. For this purpose, I really recommend to modified Figure 14 including each transect and the photo with the main characteristics species or communities inhabiting the different geological structures within transects.

Authors' response: We accept that some more specific details can be given regarding the substrates that the faunas were inhabiting, so we can elaborate on these details in this section. However, we do not believe that modifying fig 14 is beneficial or necessary for this study, which would lead to the repetition of images in the new figure combining figs 3 and 4 as requested by this reviewer. Furthermore, given the sporadic distribution and low abundance of the faunas at the dive sites, the resultant image would tend to portray similar communities, rather than the diversity of faunas observed during this expedition, the latter being most typically represented in publications of this type (see references in manuscript) and also being more informative given they represent the first visual record of these inhabitants. Please also see our overall comments above (General Query) regarding the limitations of this study and our response relating to canyon-faunal relationships as raised by the reviewer.

Manuscript changes: Add further details in this section regarding the canyon topography and substrates associated with particular faunas, and the potential influence of the hydrodynamics on fauna in the uppermost intermediate waters. We don't believe that

we can further expand on details or inferences regarding potential canyon controls on faunal distributions given the scope of this study (see above). We have not modified Fig 14 to avoid repetition of figures and images, and because we wish to show a representation of the variety of fauna we observed inhabiting the canyon, which is the typical approach taken in other studies (including references cited by the reviewer). However, to fulfil the reviewer's request to illustrate key fauna observed at each site, we propose to include a new figure that shows a map of the canyon, the dive sites, and symbols representing key taxa observed during each dive.

Reviewer's comment: Figure 2. I suggest deleting Figure 2 (not relevant data). Also, add the sampling point in Figure 1B. Additionally, I believe that the quality of Figure 1 can be improved. If possible, try to follow similar format within the Figures (i.e. regarding the position of the legend, the names of the categories, etc). Increase the size font of lat/long (not legible). I suggest including also north in the map.

Authors' response: We agree that Fig 2 can be made redundant and to the changes to Fig 1.

Manuscript changes: Delete Fig 2. Revise Fig 1, not only by increasing font size, adding site labels, and north orientation as requested. We would also replace Fig 1a with a more relevant image, being a more informative map of the location of the canyon and its relationship to the continental shelf and SW Australian coast (i.e. disconnected), which is a very important feature discussed in the text.

Reviewer's comment: Figure 3 and 4. I would merge Fig 3 and 4, makes it easy for readers to quickly understand the dives characteristics (e. g. Figure map and 2 photos of each dive)

Authors' response: We accept the reviewer's request but nevertheless emphasize that the disciplines of geology and bathymetry are somewhat different, and that the lithologies and sedimentary structures pictured in the combined new figure are not necessarily representative of each dive site (impossible to achieve with x2 images per dive)

or their geomorphological characteristics (as shown by bathymetry maps), and instead highlight some important geological features which are discussed in the text.

Manuscript changes: Merge figs 3 and 4. The lithology images removed from the original figure 4 will be integrated into the Supplementary file figures.

Reviewer's comment: Supplement Figures S1 and S2. It almost repeated from Figure 4. I would delete it. No new information is provided.

Authors' response: We disagree with the reviewer because these are the first images of this canyon system, and are therefore important to report. These various images show the variety of lithologies and sedimentary features observed throughout each dive which and are discussed in the text.

Manuscript changes: We have retained these Supplementary file figures.

Minor comments:

Reviewer's comment: Pag 1. Line 21. Changes CTD profiling for chemical Authors' response: Agree Manuscript changes: Change as requested

Reviewer's comment: Pag 3. Line. 1 (should be line 9). Don't think sub-heading is necessary for Introduction. Authors' response: We disagree with the reviewer. Manuscript changes: Retain sub-heading (no change)

Reviewer's comment: Pag5. Line 29. 10 subsamples from each site, is correct? Please clarify. Authors' response: Site A n=9, B n=9, E n=11. These specifics can be found in data links. Manuscript changes: Clarified as requested

Reviewer's comment: Pag 6. Line 1. Changes to "Vienna Standard Mean Ocean Water (VSMOW)" Authors' response: Agree Manuscript changes: Change as requested

Reviewer's comment: Pag 7. Line 29. 2.5 Add Stronium (Sr) isotope dating Authors' response: Agree Manuscript changes: Change as requested

Reviewer's comment: Pag 8. Line 8. Changes Strontium for Sr Authors' response: Agree Manuscript changes: Change as requested

Reviewer's comment: Pag.14 Line 31: Include; deeper depths Authors' response: We disagree, as the text as currently expressed is correct. Manuscript changes: No change

Reviewer's comment: Pag.14. Line 28. Delete "s" from FIGs Authors' response: Agree Manuscript changes: Change as requested

Reviewer's comment: Pag15-Line 1: Changes the sentences to "formation, suggesting that this material is most likely pelagic in origin. Authors' response: Agree Manuscript changes: Change as requested

Reviewer's comment: Figure legend 14. Acesta Italicize Authors' response: Agree Manuscript changes: Change as requested

Reviewer's comment: Pg 17. Line 25. Changes: "mobile megabenthos. Demersal.." Line 27 Changes Sea urchins Authors' response: Agree Manuscript changes: Change as requested

Reviewer's comment: Pag.16. Line 25. Changes; deep-sea fauna Authors' response: Agree Manuscript changes: Change as requested

Reviewer's comment: Figure 6. - I did not found the Small white squares representing the location of moorings in the Figure. Please include them. Please also include the north in the figure. Authors' response: Agree Manuscript changes: Change as requested

Reviewer's comment: Figure 7. Increase the size of the letters of the Figures and axis font. Authors' response: Agree Manuscript changes: Change as requested

Reviewer's comment: Figure 8. Include the north. Include in the legend; Small white squares represent the location of moorings (current meters and thermistor chains). Authors' response: Agree Manuscript changes: Change as requested

Reviewer's comment: Figure 9. The legend and colors of Figure 9A is not legible, increase the size Authors' response: Agree Manuscript changes: Change as requested

Reviewer's comment: Figure 13. Rename it appears as Fig 12. Authors' response: Agree Manuscript changes: Change as requested

———————————